



# Hidden vortices: Near-equatorial low-oxygen extremes driven by high-baroclinic-mode vortices

Florian Schütte[1,2], Johannes Hahn[3], Ivy Frenger[1],

Arne Bendinger[4], Fehmi Dilmahamod[1], Marco Schulz[1], Peter Brandt[1,2]

[1]GEOMAR Helmholtz Centre for Ocean Research Kiel, Kiel, Germany

[2]Christian-Albrechts-University, Kiel, Germany

[3]Bundesamt für Seeschifffahrt und Hydrographie, Hamburg, Germany

[4]Laboratoire d'Océanographie Physique et Spatiale, University Brest,
CNRS, Ifremer, IRD, IUEM, Brest, France

Corresponding Author: Florian Schütte (fschuette@geomar.de)

**Keywords:** subsurface low-oxygen/hypoxic patches, high-baroclinic mode vorticies, subsurface coherend vorticies, submesoscale eddies, low-latitudes / tropical / near-equatorial





**Abstract**
Long-term time series of dissolved oxygen (DO) measurements from the upper 500 m depth
of the eastern tropical North Atlantic (ETNA), collected over a period of up to 15 years at three
different mooring sites, reveal recurring extreme low-oxygen events lasting for several weeks.
Similarly, observations from 15 individual meridional ship sections between 6°N and 12°N
along 23°W show DO concentrations far below 60 μmol kg⁻¹ in the upper 200 m - significantly
lower than the climatological values at this depth (>80 μmol kg⁻¹). Two-third of these low-
oxygen events could be related with high-baroclinic-mode vorticies (HBVs) with their cores
located well below the mixed layer. Despite the energetic equatorial circulation and the
expected dominance of wave-like structures in the near-equatorial region, these HBVs persist
as relatively long-lived and coherent features. Based on moored and shipboard observations
from the ETNA, and supported by an eddy-resolving ocean-biogeochemistry model, we
characterize their dynamics and DO distribution. Observed water mass properties and model
analyses suggest that most HBVs originate from the eastern boundary and can persist for
more than six months. As they propagate westward into regions of higher potential vorticity
(PV), anticyclonic HBVs with low-PV cores remain more effectively isolated and have longer
lifespans compared to cyclonic HBVs with high-PV core. The vertical structure of the dominant
anticyclonic HBVs corresponds to baroclinic modes 4-10, with associated Rossby radii ranging
from 34 km to 13 km, respectively. This is consistent with observed eddy sizes and is well
below the corresponding 1st baroclinic Rossby radius of deformation (> 100 km). Since none
of the observed HBVs exhibit a surface signature, a substantial portion of the near-equatorial
eddy field may remain undetected by satellites, yet still exert significant influence on ocean
ecosystems and biogeochemical cycles.



## 1. Introduction

Dissolved oxygen (DO) concentration is a key component of marine ecosystems, shaping biodiversity, biogeochemical cycles, and the survival of pelagic species. From long-term moored observations in the open Eastern Tropical North Atlantic (ETNA) near the equator (latitudes <12°N), we repeatedly observe short-lived extreme low-oxygen events in the subsurface, well below the mixed layer. This DO variability is likely driven by small-scale vortices, which is unexpected, as theory suggests that wave-like structures should dominate at these latitudes (Eden, 2007). In this study, we combine moored time series, repeated ship transects, and an eddy-resolving biogeochemical model to investigate these small-scale processes below the mixed layer in the tropical Atlantic. This integrated approach allows us to characterize their structure, variability, and strong influence on DO distribution, with potential implications for marine ecosystems and biogeochemical cycles.

Extreme events of low DO in isolated cores of large coherent mesoscale eddies have become a well-studied phenomenon of the Atlantic and Pacific eastern boundary upwelling systems (e.g. Stramma et al. (2013); Karstensen et al. (2015); Schütte et al. (2016b); Frenger et al. (2018)). A strong isolation and the longevity of the eddy for at least several months favor a DO depleted eddy core. The DO depleted core results from (i) trapped water, which is transported westward within the eddy core from a region of initially low DO, typically from the eastern boundary (dynamic effect) and (ii) enhanced DO consumption (production effect) due to a biologically high productive regime above the eddy core (McGillicuddy, 2016). The latter is associated with high phytoplankton productivity, which leads to enhanced respiration and reduction of DO beneath the mixed layer directly in the isolated core reaching down to about 200 m (Karstensen et al., 2017). Respiration rates in the eddy's interior were found to be substantially increased, with 3 to 5 times the values of ambient conditions for the tropical North Atlantic, e.g. subsurface intensified anticyclonic eddies (subsurface ACEs): $0.19 \pm 0.08$ µmol kg$^{-1}$ d$^{-1}$ and surface intensified cyclonic eddies (CEs): $0.10 \pm 0.12$ µmol kg$^{-1}$ d$^{-1}$ (Schütte et al., 2016b). The increased respiration within these isolated mesoscale eddies may result in anoxic conditions (< 5 µmol kg$^{-1}$) in the otherwise hypoxic (60 µmol kg$^{-1}$) ETNA. Such eddies have a severe impact on biogeochemical processes and organisms (Fiedler et al., 2016; Hauss et al., 2016; Löscher et al., 2015). Moreover, it is suggested that the increased DO consumption within the isolated mesoscale eddy cores promote the formation and existence of a broad-scale shallow DO minimum zone (sOMZ) at about 80 m (Schütte et al., 2016b), that is most pronounced off the nutrient-rich Mauritanian upwelling system in the ETNA, located between 15° and 23°N (Karstensen et al., 2008; Brandt et al., 2015) (Fig. 1). For such low DO extremes to develop highly isolated eddies must form and propagate over a relatively long period through regions of relatively low dynamical activity, e.g. as stated in the





mentioned literature north of 12°N in the eastern Atlantic and Pacific oceans.
The occurrence of DO depleted long-lived coherent eddies in near-equatorial waters (< 12°) is
not intuitive and contrasts theoretical considerations of equatorial dynamics, which suggest a
dominace of anisotropic wave like structures (Eden, 2007). However, several extreme low DO
events have been observed in the ETNA at latitudes between 6° and 12°N (Brandt et al. 2015),
where Christiansen et al. (2018) associated one of these events (at 8°N, 23°W) with a
subsurface ACE. These eddies are expected to be less isolated and shorter-lived compared
to eddies poleward of these low latitudes. The first baroclinic Rossby radius of deformation
($R_{d,1}$), which is a characteristic threshold size of a dynamical regime to be in a geostrophic
balance on meso- and larger scales, strongly increases towards the equator (Chelton et al.,
1998). Global eddy studies, mainly based on altimeter sea surface height data, show a strong
equatorward decrease of long-lived (> 35 days) eddies (Chaigneau et al., 2009; Chelton et al.,
2011). Less isolated eddies more readily entrain DO from surrounding waters, while short-lived
eddies do not persist long enough to substantially deplete DO in their cores. Both factors inhibit
the development of a low DO extreme within eddy cores. Additionally, the equatorial region -
compared to the eastern parts of the oceans north of 12°N - is highly dynamic. It features the
energetic equatorial zonal current system with associated instabilities as well as various wave
phenomena (e.g. Urbano et al. 2006; Pena-Izquirdo et al., 2015; Calil et al., 2023; Köhn et al.
2024). Nevertheless, our observations occasionally reveal DO values significantly below the
climatological value (Fig. 1).

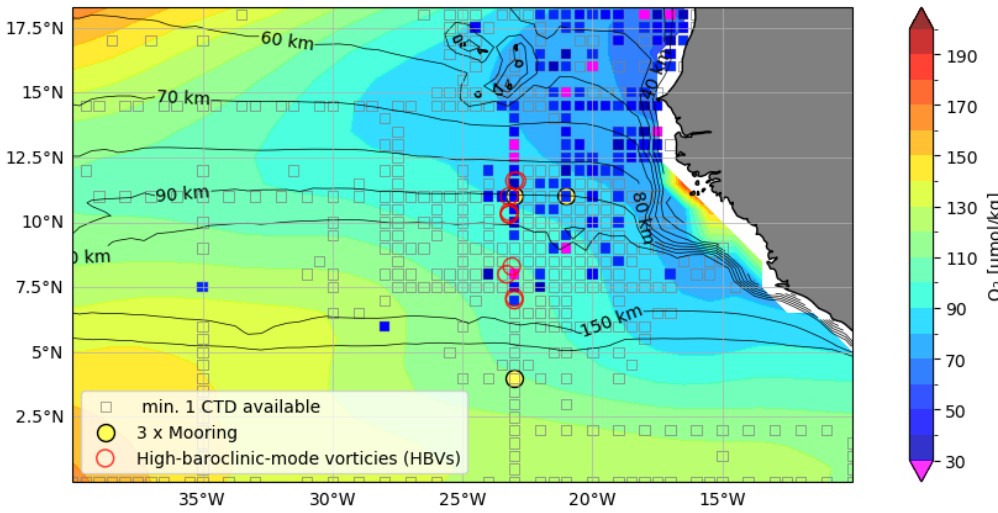

Figure 1: Map of the eastern tropical North Atlantic. Shaded are minimum DO values in the
upper 200 m of the climatological DO distribution from the World Ocean Atlas 2023. The small
squared boxes indicate regions of 0.5 degree boxes for which at least one CTD station is



available. These boxes are colored with their minimum DO concentration in the upper 200m
(from multiple CTDs, if available) only if the minimum DO concentration is less than 60 µmol/kg.
Red circles suggest the occurrence of high-baroclinic mode vorticies as analyzed in detail in
the manuscript. The yellow points mark the positions of the moorings analyzed in the
manuscript. The black contour indicates the first baroclinic Rossby radius of the deformation
(in km), calculated from the World Ocean Atlas data, following Chelton et al., (1998).

While mesoscale eddies of the first baroclinic mode can hardly exist at near-equatorial
latitudes, smaller-scale eddies might. In the following, we refer to these smaller eddies, which
still exhibit similar dynamics to mesoscale eddies - i.e., they are dominantly in geostrophic
balance – as high-baroclinic mode vortices (HBVs). They have radii below the first baroclinic
Rossby radius, $R_{d,1}$, and baroclinic modes larger than one (D'Asaro, 1988; McCoy et al., 2020,
McWilliams et al 1985, 2016). These HBVs, often referred to in the literature as subsurface or
submesoscale coherent vortices, are observed to have isolated cores and can therefore advect
tracers (Gula et al., 2019). Due to their small spatial scales and since they often appear at
subsurface depth, these eddies are not necessarily detectable from satellite observations
(McCoy et al., 2020). HBVs are not typically known to persist for extended durations in near-
equatorial waters. However, here we provide evidence that HBVs may serve as a potential
mechanism driving the observed low-oxygen extremes at these low latitudes. When linked to
biogeochemical anomalies, HBVs can play a crucial role in biogeochemical cycles and marine
ecosystems. However, ocean models  are often submesoscale-permitting only, such that
submesoscale processes are not fully resolved, particularly with increasing distance from the
equator. Understanding the frequency and behavior of HBVs is essential for understanding
tracer distributions, developing effective parameterizations and improving model accuracy.
In this study, we identify the characteristics, origin and temporal evolution of low-oxygen
extremes in the upper 200 m of the tropical Atlantic Ocean and discuss the role of HBVs in
driving these DO deficient zones. We use a comprehensive data set of in situ moored and
shipboard observations combined with an actively eddying ocean-biogeochemistry model
(respective data and methods introduced in *section 2* and *3*) in order to investigate the
frequency distribution and magnitude of these events (*section 4.1*). We show that the low-
oxygen events are by the majority related to subsurface intensified HBVs (*section 4.2*), both
anticyclonic and cyclonic. We derive the demography of these structures from an eddy
detection algorithm applied to in-situ observations (*section 4.3*) and a vertical baroclinic mode
analysis (*section 4.4*). The core water of the HBVs is analysed in *section 4.5* and the origin
and temporal evolution of the HBVs based in model simulations is shown in *section 4.6.* We
give a detailed discussion in *section 5* and provide a summary in *section 6*.



## 2. Data

Data from moored, shipboard and satellite observations, climatological data as well as the
output of an actively eddying ocean-biogeochemistry model from the tropical North Atlantic
were used within this study as described in the following.

### 2.1. Moored observations

Multi-year moored observations from three different locations 11°N, 21°W; 11°N 23°W and 4°N
23°W were used in this manuscript (Fig. 1). The mooring at 11°N, 21°W was equipped with
DO (AADI Aanderaa optodes of model types 3830 and 4330) and CTD (Conductivity,
temperature, depth) sensors (Sea-Bird SBE37 microcats) which were attached next to each
other on the mooring cable between 2012 to 2018. Eight of these optode/microcat
combinations were installed evenly distributed in the depth range between 100 to 800 m,
delivering multi-year time series of temperature, salinity and DO with a temporal resolution of
up to 5 min. At 800 m depth, an upward looking Acoustic Doppler Current Profiler (ADCP) was
installed to record velocity in the depth range between about 60 and 800 m. During the 2$^{nd}$
deployment period (May 2014 to Sep 2015), no velocity observations were available due to a
failure of the ADCP. Before and after a deployment period, optodes and microcats were
calibrated against CTD-O measurements during CTD casts and onboard lab measurements
as described in Hahn et al. (2014) and Hahn et al. (2017). The correction against reference
measurements, thereby considering potential sensor drifts (Bittig et al., 2018), allowed best
data quality and yielded average root mean square calibration errors of 0.003°C, 0.006 and
3 µmol kg$^{-1}$ for temperature, salinity and DO, resepectively. Only quality controlled data that
was flagged good was used for further analysis. ADCP measurements were quality controlled
against a percent good criterion (20% threshold) and were checked for plausibility and evident
outliers due to surface reflection. ADCP bin depths were corrected using a mean sound speed
profile following the approach by Shcherbina et al. (2005). This mooring is used to study
hydrographic, DO and velocity temporal variability (on daily to intraseasonal time scales)
related to low-oxygen extreme events. The other moorings at 11°N 23°W and 4°N 23°W are
part of the prediction and research moored array in the tropical Atlantic (PIRATA), which were
equipped with DO (AADI Aanderaa optodes of model types 3830 and 4330) sensors at 300 m
and 500 m depth from 2009 to 2024. At 11°N, 23°W additionally an DO sensor at 80 m depth
was installed between 2017 to 2024. The DO sensors deliver hourly data and are calibrated
and processed in the same way as described above.





### 2.2. Shipboard observations


Hydrographic and DO data was obtained from CTD-O casts, that were carried out during a
large number of research cruises to the tropical North Atlantic between 2006 and 2022. In the
region 6°-12°N and 30°-18°W, 976 profiles were recorded during 24 cruises mainly covering
the upper 1300 m of the water column. Two independently working systems of temperature-
conductivity-pressure-oxygen sensors were used, that allowed to identify spurious sensor
data. Salinity and DO readings were calibrated against values from water samples, that were
taken during the majority of CTD-O profiles of each individual cruise and that were measured
onboard with salinometry and Winkler titration, respectively. For a single cruise, data accuracy
was generally better than 0.002°C, 0.002 and 2 µmol kg for temperature, salinity and DO,
respectively.
The majority of research cruises covered the 23°W meridian in the tropical North Atlantic. They
captured several low-oxygen extreme events in the latitude range between 6° and 12°N (Fig.
1). We made use of these CTD-O observations that were mostly carried out at a meridional
resolution of 0.5°, corresponding to 55 km, in order to investigate the spatial distribution of the
low oxygen extremes. Horizontal velocity data were additionally acquired continuously along
the cruise track with vessel-mounted Acoustic Doppler Current profilers (vmADCPs). The
typical vmADCP operating frequency was 75 kHz, where 1-hour averaged data has an
accuracy of better than 2-4 cm s$^{-1}$ (Fischer et al., 2003). During one cruise, a vmADCP system
with 150 kHz operating frequency was used and we expanded this data set with data from a
lowered ADCP (lADCP), that was attached to the CTD rosette and measured velocity profiles
at CTD-O cast positions. Single velocity profiles from lADCP had an accuracy of better than
5 cm s$^{-1}$ (Visbeck, 2002). The horizontal velocity observations from all 23°W ship sections
covered the depth range of the upper 300 m, the depth where the extreme low-oxygen occur
and thus coinciding with our target depth range.
For each 23°W ship section hydrography, DO and velocity were mapped onto a regular depth-
latitude grid (resolution of 10 m and 0.05°) using a Gaussian interpolation scheme with vertical
and horizontal influence (cutoff) radii of 10 m (20 m) and 0.05° (0.1°), respectively (for details
see Brandt et al. (2010). This is done to plot the average section along 23°W in order to
compare it with the model data and assess the model performance.

### 2.3. Satellite observations


Sea level anomaly (SLA) and surface geostrophic velocity derived from satellite altimetry
products were used in this study to identify the surface signatures of eddies. The multimission
Data Unification and Altimeter Combination System (DUACS) product in delayed time and





daily resolution with all satellite missions available at a given time is used. It has a spatial
resolution of 0.25° and is provided by Marine Copernicus (https://doi.org/10.48670/moi-

210    00148).

**2.4. Climatological data**
Gridded climatological hydrography and DO from the World Ocean Atlas 2023 (WOA23)
(described e.g. in Reagan et al. 2024) was used as a reference data set throughout this study.
In addition, the monthly, isopycnal and mixed-layer ocean climatology (MIMOC) (Schmidtko et
al., 2013) was used. For more details see section *6 Data availability*.
**2.5. Coupled ocean-biogeochemistry model**
We used the output of the GFDL climate model with an eddy-rich ocean, CM2.6 (Delworth et
al., 2012; Griffies et al., 2015) to further understand the origin and development of low-oxygen
extremes in the tropical ocean. CM2.6 has a nominal ocean resolution of 0.1 degrees and an
atmosphere with approximately 50 km resolution. For computational efficiency, marine
biogeochemistry is represented by the simple biogeochemical model MiniBLING (Galbraith et
al., 2015). MiniBLING was run with the three prognostic tracers dissolved inorganic carbon,
phosphate and DO. Organic carbon (biomass) is treated diagnostically and is not advected in
the model. Despite its simplicity, MiniBLING has been shown to perform comparably well to
more complex marine biogeochemical models in simulating marine biogeochemistry and its
sensitivity to climate (Galbraith et al., 2015). Moreover, the small number of tracers was not
just a limitation but a key factor that made it possible to run a simulation with a mesoscale-rich
ocean.
The results we show here stem from a simulation with preindustrial atmospheric carbon dioxide
levels that has been run for 200 years, with marine biogeochemistry coupled at year 48. The
model has been spun up from rest with initial conditions from WOA09 (Locarnini et al., 2010;
Antonov et al., 2010; Garcia et al., 2010a; Garcia et al., 2010b) and Global Ocean Data
Analysis Project (GLODAP) (Key et al., 2004). For more details on the model set up and a
general evaluation of the model we refer to Griffies et al. (2015) and Dufour et al. (2015). Here,
we used five daily model outputs of the last 20 years of the simulation. A brief evaluation of the
model performance for the northern hemispheric Atlantic DO conditions, the focus of our study,
is given in the following.



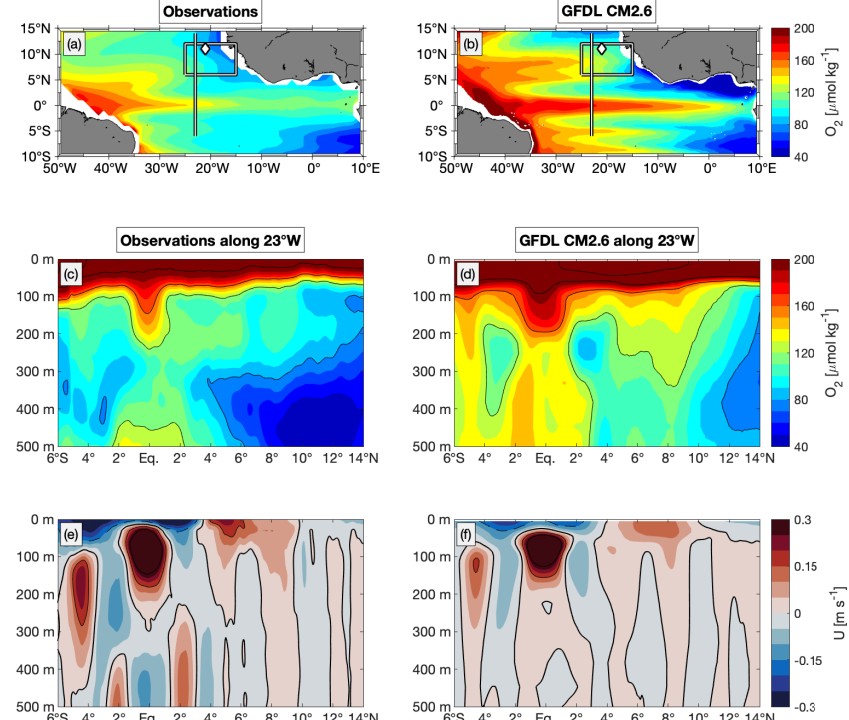


Figure 2: Observation-model comparison of the minimum DO between 0 and 200 m of the time-average distribution from (a) MIMOC and (b) from last 20 years of GFDL CM2.6 model. Latitude-depth section, 0-500 m along 23°W, of mean DO from (c) repeat ship sections and (d) last 20 years of GFDL CM2.6 model. (e) and (f) are similar to (c) and (d), but mean zonal velocity is shown. The box in (a) and (b) illustrates the area of interest in this study; the line denotes the 23°W section that is shown in subpanels (c) to (f). This section has been surveyed by 15 individual shipboard observations that are used in this study for the latitude range 6°N-12°N. Diamond marks the mooring position (21°W, 11°N), where data used in this study were taken.

The distribution of the minimum DO between 0 and 200 m taken from the time averaged over
the last 20 years of the GFDL CM2.6 model output (Fig. 2b), shows similar large-scale patterns
as the same corresponding distribution taken from the MIMOC climatology (Fig. 2a): well
oxygenated western boundary region, decreasing DO values toward east with off-equatorial
OMZs on both sides of the equator showing minimal values at the eastern boundary. The
simulated distribution has higher DO concentration at the western boundary and in the interior
basin, and partly lower values at the eastern boundary compared to the climatological
distribution from observations, which is particularly the case in the Gulf of Guinea region. In
the interior basin, meridionally alternating bands of oxygen-poor and oxygen-rich water, that
are associated with shallow east- and westward current bands, are pronounced in GFDL
CM2.6, albeit more intensified.





In the ETNA, the average DO distribution along 23°W in GFDL CM2.6 (Fig. 2d) shows a
notable mismatch with observations (Fig. 2c). While observations from repeat ship sections
reveal two distinct OMZ layers - a shallow OMZ above 200 m and a deeper OMZ at 300 - 700
m - the model instead simulates only a single OMZ spanning 100 - 600 m. This bias is also
present in other coupled ocean circulation biogeochemistry models (e.g. Duteil et al., 2014).
Further differences appear south of the equator, where the observed OMZ is absent in the
model along 23°W. Instead, GFDL CM2.6 simulates lower DO levels between 2°- 4°N at
depths below 150 m compared to observations. The corresponding section of zonal velocity
(Fig. 2f) indicates that the model represents upper-ocean currents (above 200 m) well when
compared to observations (Fig. 2e). However, below 200 m in the equatorial region (5°S -5°N),
zonal currents are considerably weaker and partly misrepresented. North of 5°N, the velocity
structure is generally better captured.
Despite differences in spatial details and magnitudes the basic features of the DO and velocitiy
distributions are in the upper 200 m of the ETNA, and the GFDL CM2.6 model provides a
robust physical and biogeochemical background state to study the role of eddies in driving
local DO deficient zones. The model has been shown to simulate low-oxygen mesoscale
eddies at latitudes poleward of about 12° latitude (Frenger et al., 2018). Here, we use the last
20 years of this model run to study low-oxygen extreme events in the ETNA equatorward of
12°N.
**3.  Methods**
Different diagnostics have been applied in this study, that allowed us to associate low-oxygen
features with HBVs and to analyze their origin and temporal evolution. The concept of vertical
baroclinic modes (*section 3.1*) was used to characterize the vertical structure of HBVs, to
identify the dominant vertical modes, and their associated Rossby radius of deformation and
propagation speed. In *section 3.2*, we briefly present the calculation of PV, which is used as a
conservative tracer to track and to identify the isolation of different water masses. In *section
3.3*, we describe the different approaches for eddy identification from shipboard observations
and in the GFDL CM2.6 model.
**3.1. Vertical baroclinic modes and Rossby radius of deformation**
A powerful way to describe linear wave dynamics in the ocean is the decomposition into vertical
baroclinic modes (Philander, 1978). Each baroclinic mode is associated with a specific gravity
wave speed and a corresponding Rossby radius of deformation, which defines its
characteristic horizontal length scale.





### 3.1.1. Baroclinic mode decomposition

The concept of baroclinic modes is based on the linearized hydrostatic equations of motion,
which can be separated into a horizontal and a vertical component. Assuming a motionless
background state and a flat-bottomed ocean, the vertical structures are given by solving the
eigenvalue problems (Gill, 1982):

$$\frac{d^2\Psi_n(z)}{dz^2} + \frac{N^2(z)}{c_n^2}\Psi_n(z) = 0 \tag{1}$$

where $\Psi_n(z)$ describes the vertical structures of isopycnal displacement $\xi$ or vertical velocity
$w$ and $z$ is the vertical coordinate. $N(z)$ is the vertical profile of the Brunt-Väisälä frequency
and $c_n$ the gravity wave speed for mode $n \in \mathbb{N}$. For the eigenvalue problem (1), we use
boundary conditions with a free surface and a flat bottom (Gill, 1982), which are given as

$$\Psi_n = \frac{c_n^2}{g}\frac{d\Psi_n}{dz} \text{, at } z = 0 \quad \text{and} \quad \Psi_n = 0 \text{, at } z = -H \tag{2}$$

where $H$ is the ocean depth and $g$ the gravitational acceleration. For a continuously stratified
ocean, the number of solutions depends on the vertical resolution of the data used.  Any
perturbances can be described as a superposition of orthogonal vertical baroclinic modes ($n =$
$1, 2, 3, ...$). Amplitudes of vertical structure functions are normalized such that

$$\int_{-H}^{0} \Psi_n \Psi_m dz = \delta_{nm} H$$

where $\delta_{nm}$ is the Kronecker delta and $n, m$ the modes. The gravity wave speed is related to
the Rossby radius of deformation, that can be calculated for the off-equatorial regions
(poleward of 5°S and 5°N) as

$$R_{d,n} = \frac{c_n}{|f|} \tag{3}$$

(Gill, 1982 or Chelton, 1998), where $R_{d,n}$ is the Rossby radius of deformation for the $n$-th
vertical baroclinic mode, and $f$ is the Coriolis parameter.



### 3.1.2. Calculation of vertical baroclinic modes and modal decomposition

The main goal is to decompose any disturbed state into the set of orthogonal baroclinic modes
that solve (1). Each hydrographic profile from an individual CTD-O profile can be considered
as a perturbation from the mean state. The mean state distribution was derived from the 3-D
climatological hydrographic field (cf. Chelton et al. (1998)) that is given by the World Ocean
Atlas (*section 2.4*). Given the corresponding density profile, we calculated the isopycnal
displacement $\xi(z)$ by

$$\xi(z) = \frac{\rho'(z) \cdot g}{\rho_0 \cdot N^2} \tag{4}$$

with $\rho'(z) = \rho(z) - \rho_{ref}(z)$, $\rho_0 = 1025$ kg/m³ a constant reference density and $\rho_{ref}$ being the
undisturbed profile of potential density (here taken as the climatological density profile from
the World Ocean Atlas – see also Vic et al. 2021 for more details on the method used). The
isopycnal displacement of the disturbed state can be described as a superposition of the
orthogonal set of vertical baroclinic modes for displacement, i.e.

$$\xi(z) = \sum_{n=1}^{K} x_n \Psi_n(z) \tag{5}$$

Here, $K \to \infty$ expresses the exact solution with an infinite number of vertical modes for a
continuously stratified ocean. The expansion coefficients $x_n$ are the modal amplitudes. The
modal amplitudes are obtained by projecting the observed fields onto the structure functions
computed from the World Ocean Atlas. The projection is preferred over resolving a least-
square problem, which sometimes leads to unrealistic modal amplitudes into the high modes
(Vic et al. 2023). The modal amplitudes $x_n$ are calculated via a scalar product:

$$x_n = \int_{-980}^{-30} \psi_n(z) \cdot \zeta_{CTD}(z) \, dz \tag{6}$$

These amplitudes are then normalized by dividing with $\int_{-980}^{-30} \psi_n(z)^2 \, dz$. This analysis is
restricted to the depth range from 30 m to 980 m in order to exclude the surface mixed layer
while retaining the majority of available profiles along 23°W. The barotropic mode assumed to
be zero. A vertical resolution of 10 m is used, with both the CTD profiles and World Ocean
Atlas data interpolated accordingly. After computing the contribution of one mode, it is
subtracted from the displacement profile: $\xi'(z) = \xi(z) - x_n \Psi_n(z)$ and the procedure is





repeated for the next mode. This recursive removal reduces cross-talk between modes caused
by the limited vertical resolution and incomplete depth coverage. Since the order of mode
extraction may influence the result, the decomposition is repeated $M = 100$ times with random
permutations of modes $n = 1\ to\ n = 20$, and the final modal amplitudes are calculated as the
mean over all realizations, with associated standard errors.

### 3.2. Potential vorticity and Rossby number

Subsurface eddies exhibit signatures of high or low potential vorticity (PV), depending on their
stratification anomaly and rotation direction (D'Asaro, 1988; McWilliams, 1985; Molemaker et
al., 2015). In the absence of mixing, PV is a conserved quantity and serves as an effective
tracer to differentiate water masses and track eddy pathways.
We refer to Ertels PV (Gill, 1982), being one of the most complete formulations for PV
conservation, and take its vertical approximation (see e.g. Thomsen et al. (2016)), which is
given by

$$Q = (\zeta_z + f) \cdot N^2 \tag{7}$$

where $\zeta_z = \frac{\partial v}{\partial x} - \frac{\partial u}{\partial y}$ is the vertical component of the relative vorticity and $f$ is the Coriolis
parameter. The term $\zeta_z + f$ represents the absolute vorticity. The approximation given by (7)
is valid in case of nearly horizontally orientated isopycnal surfaces (Thomsen et al. 2016).
Counter-clockwise and clockwise rotating eddies correspond to positive and negative relative
vorticity, respectively. In the northern hemisphere, anticyclonic eddies rotate clockwise and
have negative relative vorticity (vice versa for the southern hemisphere, which is not further
considered throughout this study).
In the case of geostrophic balance, the Rossby number

$$Ro = \frac{U}{Lf} = \left|\frac{\zeta_z}{f}\right| \tag{8}$$

where $U$ is characteristic velocity and $L$ is characteristic length scale, is smaller than one and
PV is always positive. PV can be reduced by either a reduction of $N^2$ (weakened stratification)
or by a gain of anticyclonic relative vorticity (D'Asaro, 1988). The explanation also applies vice
versa, i.e. PV can be increased by a strengthening in stratification or a gain of cyclonic relative



vorticity. The Rossby number becomes larger than one for submesoscale dynamics in the
ageostrophic range.

### 3.3. Eddy identification algorithms

#### 3.3.1. Eddy identification from shipboard observations

Horizontal velocity data from the vmADCP system (see *section 2.1*) is used to detect eddies
along the 23°W meridian between 6°N and 12°N. This methodology is based on an idealized
eddy solution, known as Rankine vortex characterized by solid-body rotation in its inner core,
i.e., a linear increase of velocity with increasing distance from the eddy center. We do so
through the conversion from Cartesian into cylindrical coordinates in areas that are suspected
to cross eddies. Every point in the horizontal plane is defined by the radial distance, $r$, to the
origin (eddy center) and the azimuthal angle, $\theta$, i.e.,:
$$v_r = u \cos \theta + v \sin \theta \tag{9}$$

$$v_\theta = -u \sin \theta + v \cos \theta \tag{10}$$

where $v_r$ and $v_\theta$ are the radial and azimuthal velocities, respectively. Following Castelao and
Johns (2011) and Castelao et al. (2013) the optimal eddy center is found by minimizing $v_r$
(maximizing $v_\theta$ ) via a non-linear least-squares Gauss-Newton algorithm.
$$|v| = -u \sin(\theta) + v \cos(\theta) + \epsilon \tag{11}$$

$$\theta = \arctan(y_r / x_r) \tag{12}$$

$$y_r = y - y_c \tag{13}$$

$$x_r = x - x_c , \tag{14}$$

where $(x,y)$ are the position vectors of the velocity samples, and $(x_c ,y_c)$ the eddy center
location. The residual $\epsilon$ represents the radial velocity to be minimized. This methodology
assumes a radially axisymmetric and non-translating vortex. The optimal eddy center allows
us to determine an azimuthal velocity structure, which in turn provides an estimate for the
speed-based eddy radius, i.e. the radius where the azimuthal velocity is maximum while
separating the inner core from the outer ring.

#### 3.3.2. Eddy identification in GFDL CM2.6 model

From the GFDL CM2.6 model, we analyzed the position and trajectory of an individual
simulated eddy that was representative in terms of its westward propagation in low latitude





waters and its associated DO minimum. The horizontal eddy center at each model time step
was identified by locating the maximum of the streamfunction within a predefined 3° x 3°
longitude-latitude box centered on the eddy on a defined isopycnal surface. Once the position
was found, we searched for the DO minimum around the eddy center within a 0.8° x 0.8°
horizontal box. On average, the deviation between the two positions was about 20 km.
Additionally, for each time step, we extracted the following variables: DO and salinity on
isopycnal surface 26.5 kg m$^{-3}$, PV on the isopycnal surface 26.6 kg m$^{-3}$ (corresponding to the
isopycnal layer of minimum PV, see Fig. 4m), phosphate, and biomass integrated over the top
100 m, and particulate organic phosphorus at 100 m (to identify the downward flux of organic
matter to the eddy core). To assess eddy anomalies, we compared these variables to their
corresponding values outside the eddy (average over the area 1° to 3° longitude/latitude away
from the eddy core), and also calculated the 20-year model mean at the eddy core position.
Around the eddy position, we identified the streamline with the strongest swirl velocity and
calculated the eddy radius $R = \frac{A}{2pi}$, assuming an isotropic circular eddy, where A is the
circumferences (length of contour). The swirl velocity $U$ was calculated as the average of the
absolute value of the horizontal velocity along this contour. The propagation speed $c$ was
derived from the horizontal distance between adjacent eddy core positions (as determined by
the streamfunction criterion). To assess the isolative character of the eddy, we calculated the
isolation parameter U/c, where values greater than 1 indicate isolation of the eddy core water
from surrounding water masses.
**4. Results**
**4.1. Near-equatorial low-oxygen events: Frequency, magnitude and duration**
In all depth of the long-term DO time series from moored observations at 4°N and 11°N (both
at 23°W), recurring dips in DO levels are observed that fall significantly below the climatological
mean (Fig. 3 a, b). A low-DO extreme event is defined when DO values drop below the 10th
percentile of the respective time series. These events typically last from several days to a few
weeks and stand out clearly in the time series. They are often accompanied by a temperature
increase (Fig. 3c, d). On average, around two such events per year are observed at 4°N at
both 300 m and 500 m depth. At 11°N, about one event per year is seen at those depths, and
approximately two per year at 80 m. A similar pattern is found in the moored time series from
11°N, 21°W, where ten low-oxygen events (40–60 µmol kg$^{-1}$) were recorded between 2012
and 2018 in the upper 200 m. Each event lasted about 3 - 4 weeks.



As expected, is the DO variability and amplitude in general higher at shallower depths (e.g. 80
m), driven by stronger near-surface dynamics and the higher background DO concentrations.
Therefore the largest DO drops were typically observed at 80 - 100 m. In terms of spatial
variability, we see that the DO variability within the core of the deep oxygen minimum zone
(OMZ) at 11°N is generally lower than at 4°N.

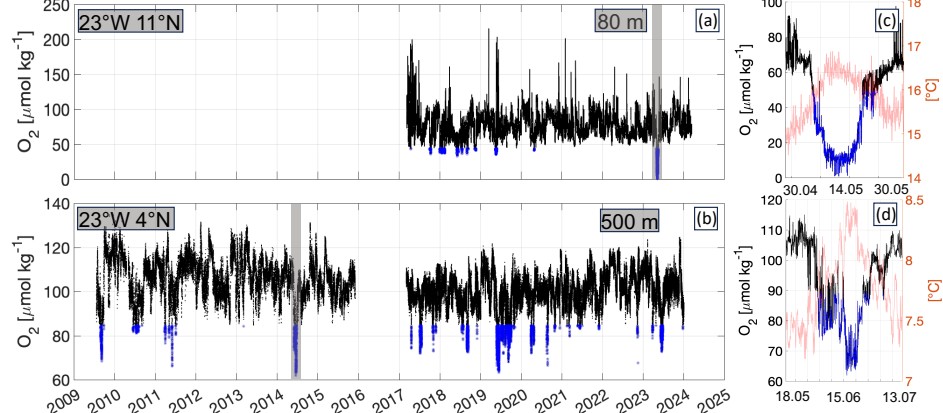


Figure 3: Time series of observed DO at (a) 80m depth at 23°W, 11°N and 500 m depth at
23°W, 4°N shown in black. The blue color represent the lowest 10th percentile of the time
series data. The grey boxes mark the timespan which is shown in (c) and (d), where the
temperature is overlaid in red.
In addition to the moored DO observations, there are multiple years of shipboard
measurements in the region. From all these shipboard observations, low-DO extremes are
identified by searching for the minimum DO concentration in the upper 200 m of every single
CTD-O profile. A low-DO extreme event was defined when DO was below a threshold of
60 µmol kg$^{-1}$, which represents the 10-percentile of all DO observations (74 of 976) in the area
6°-12°N, 30°-18°W (Fig. 4a and 4c). This threshold is more than 20 µmol kg$^{-1}$ below the
climatological DO concentration in the ETNA (Fig. 2a and 2b). Considering the absolute DO
concentration allowed us to derive a distribution of low-oxygen extremes, which is not masked
by the mean distribution. Lowest DO concentrations below 40 µmol kg$^{-1}$ (7 of 976 CTD-O
profiles) in the near equatorial region (south of 12°N) remarkably occurred not east of 21°W,
where profiles are located within a distance of 8° to the African coast, but in the "open-ocean"
region west of it (24°-21°W) (Fig. 4a). Further to the west (>24°W), lowest DO concentrations
were found again just above 40 µmol kg$^{-1}$. This is in contrast to the more coastal upwelling
region north of 12°N, where very low-DO extremes can also be observed near the coast (see
Fig. 1 or Schütte et al. 2016b). In order to scale for the different number of CTD-O profiles in



the four regions shown in Fig. 4a (5%, 9%, 61% and 25% of the profiles for the boxes 30°-
27°W, 27°-24°W, 24°-21°W, 21°-18°W), we estimated the relative distribution and calculated
the 10-percentile threshold in every box (Fig. 4c). This threshold is lowest in the open ocean
(24°-21°W), whereas the mean DO distribution is increasing from the eastern boundary
towards west. This counterintuitive distribution of low-oxygen extremes, which is against the
mean DO gradient, suggests that DO depleted water generally cannot be purely advected from
a remote region at the eastern boundary, that is poor in DO. Locally enhanced biological
activity associated with enhanced DO consumption must play a role as well.

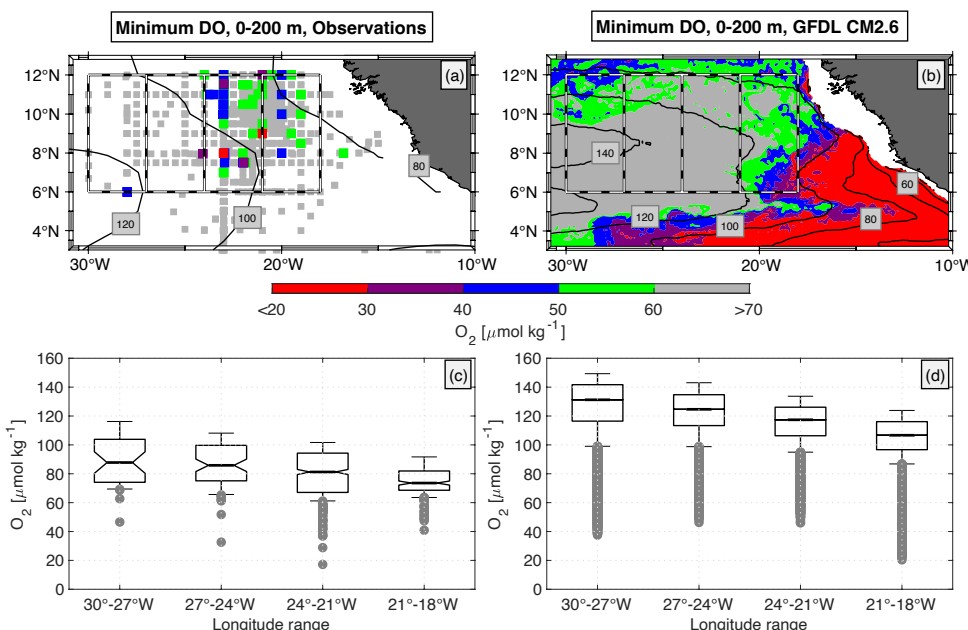


Figure 4: (a) Spatial distribution of DO profiles acquired from shipboard CTD-O observations
in the tropical North Atlantic. Colored / gray dots denote DO profiles with a minimum DO
concentration of lower / higher than 60 µmol kg$^{-1}$ in the depth range 0-200 m (red: < 30
µmol kg$^{-1}$, violet: 30-40 µmol kg$^{-1}$, blue: < 40-50 µmol kg$^{-1}$, green: < 50-60 µmol kg$^{-1}$. (b)
Horizontal distribution of DO minimum obtained in the depth range 0-200 m and from the last
20 years of GFDL CM2.6 model run. Black contour lines in (a) and (b) show 0-200 m minimum
of mean DO distribution (similar to filled contours in Fig. 2a and 2b). Dashed boxes denote
different regions of interest for boxplots shown in (c) and (d). (c) Boxplots for 0-200 m minimum
of DO profiles, that are shown for four different regions by the dashed boxes in (a). Thick line
in each boxplot denotes median and notches show 95% confidence interval. Upper and lower
whiskers denote 10% and 90% quantiles. Grey dots below the lower whiskers show 10%
lowest DO events. (d) Similar to (c), but boxplots shown for 0-200 m minimum of DO profiles
that were taken from the last 20 years of GFDL CM2.6 model run



The two events with the lowest dissolved oxygen (DO) concentrations were measured as 17 µmol kg⁻¹ by a CTD at 60 m depth at 8°N, 23°W, while concentrations even below 5 µmol kg⁻¹ were recorded by a mooring at 80 m depth at 11°N, 23°W. These two low-oxygen extremes were well below the climatological average minimum DO concentration for the whole ETNA (40 µmol kg⁻¹ in the deep OMZ, Brandt et al. (2015)). We shall note, that no CTD-O profiles were available in this data set for the eastern boundary region within about 2° longitude off the African coast.

## 4.2 Association of low-oxygen events with subsurface high-baroclinic mode vorticies

For the majority of the ship based data and for the mooring at 21°W, 11°N additional observations of hydrography, zonal and meridional velocity are available indicating the passage of anticyclonically and cyclonically rotating vortices asciated to the low-oxygen events. At the mooring position the low-oxygen events #01, #02, #03, #04 and #07 were most likely related to the passage of subsurface intensified vortices, whereof events #02, #04 and #07 were associated with anticyclonic vortices and events #01 and #03 with cyclonic vortices (Fig. 5). Note, that we explicitly refer here to the notation *vortex*, since we could not derive the vortices' radii in order to differentiate between mesoscale and submesoscale. For the anticyclonic vortices, meridional velocity was observed with maximum northward and southward flow taking place at the beginning and the end of each low-DO period. Zero crossing was observed in between at around the time, when DO was at its minimum (Fig. 5e to 5h). Corresponding time series of potential density derived from hydrographic observations, conducted next to the DO sensors, indicated a depression of isopycnal surfaces in the depth range below 100 m. Time series of velocity and potential density agree well with the dynamical understanding and passage of westward propagating eddies (van Leeuwen, 2007) through the mooring site. Zonal velocity was either small or showed maximum flow during time periods of minimum DO, depending whether the eddy has crossed the mooring site either with its core or with one of its meridional flanks. Zonal velocity vanished at the beginning and the end of each of the three events.

During events #01 and #03, that are associated with the passage of subsurface intensified cyclonic vortices, we found a depression of isopycnal surfaces above 150 m and a heave of isopycnal surfaces below ((cf. McGillicuddy (2015), denoted as eddies of type Thinny). This is associated with a maximum in stratification at about 150 m depth. The time series of zonal and meridional velocity, respectively, showed maximum values at a similar depth with a transition from westward to eastward (event #01) and southward to northward (event #03) velocities



during the time of maximum stratification. In contrast to the anticyclonic vortex events (#02, #04 and #07), the DO minima during the passage of the two cyclonic vortex events (#01 and #03) were of similar intensity at 100 and 200 m depth, with no separation from the deep OMZ at 300 m by an intermediate DO maximum. Though, during both events the minimum DO at 100 m was well below the average DO concentration that was observed for time periods without any vortex event. We shall explicitly note, that the characteristics for zonal and meridional velocity during event #01 were swapped compared to the other eddy events (#02, #03, #04 and #07). We can only speculate whether this cyclonic vortex has crossed the mooring site in a more meridionally directed pathway.

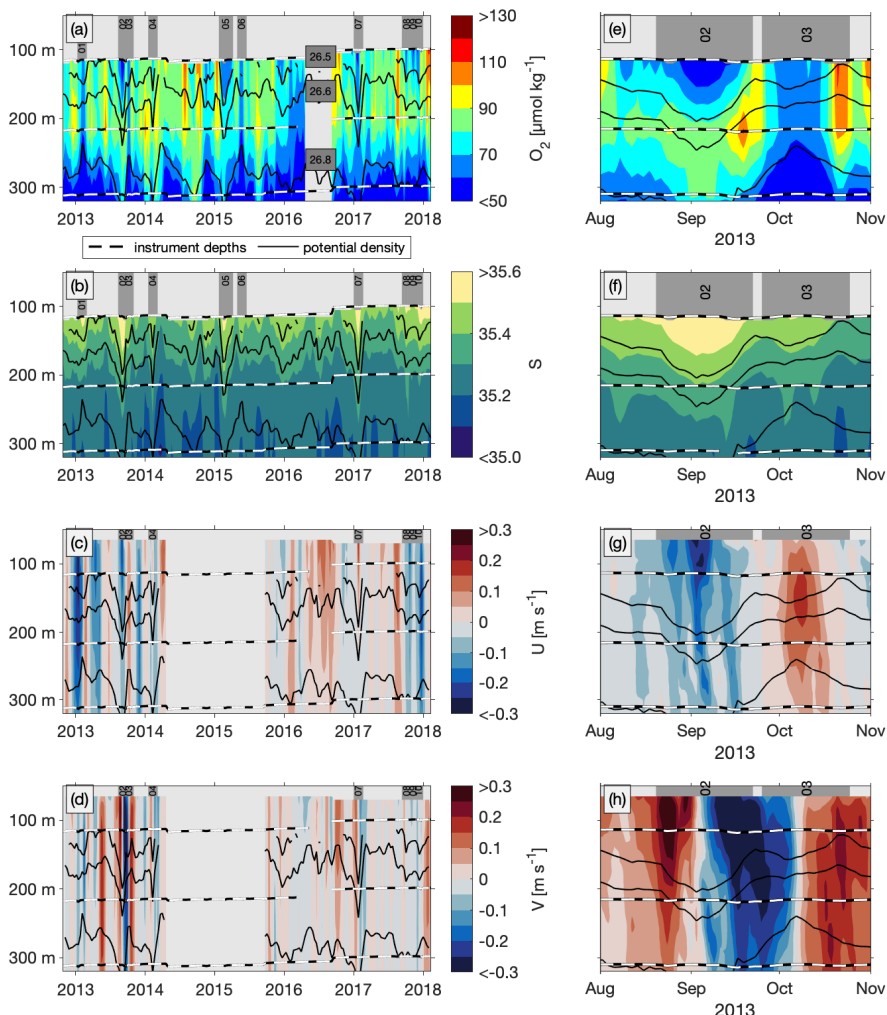

Figure 5: Time series of observed (a) DO, (b) salinity, (c) eastward and (d) northward velocity from moored observations at 11°N, 21°W in the upper 300 m as a 10-day average (color shading). Black lines denote depth of potential density surfaces 26.5, 26.6 and 26.8 kg m$^{-3}$.



Black-white dashed lines denote depths of DO sensors (in (a), (c) and (d)) and salinity sensors
(in (b)). Gray bars with numbers 01-10 in the top of these panels denote time periods of low-
DO events (#01 to #10). Note, that no velocity observations are available for low-DO events
#05 and #06. Panels (e)-(h) show corresponding 2-day averaged time series for the 90-day
time period around low-DO events #02 and #03.
The vertical structure of these vortices could not be identified for the near surface layer and
the deep ocean, since moored hydrographic and velocity observations were only available
between 100 m (60 m for velocity) and 800 m depth. This made it challenging to distinguish
among surface intensified and subsurface intensified (but at shallow depth) vortices. The most
likely subsurface intensified vortex was associated with event #02, showing extreme velocity
(both zonal and meridional) slightly below the shallowest depth of available observation
accompanied by an oxygen minimum of 39 µmol kg$^{-1}$.

### 4.3 Horizontal extent of the low-oxygen high-baroclinic mode vorticies

The ship-based data, which cover the region spatially, are significantly better suited than the
stationary moored data for assessing the spatial extent of the HBVs. Repeated meridional ship
sections between 6-12°N along 23°W, available over a distance of at least 300 km, captured
15 events with DO concentrations well below 60 µmol kg$^{-1}$ in the upper 200 m (Table 1, Fig.
6,). All DO minima were found directly below the shallow oxycline at depths between 45 and
90 m (corresponding to surfaces of potential density between $\sigma_\theta$ = 26.2 and 26.4 kg m$^{-3}$). The
meridional resolution of CTD-O measurements did not allow for a proper identification of the
meridional core position of the low-DO extremes, but their extent was found with roughly 1° in
latitude in maximum. The low-DO cores vertically extended to the isopycnal $\sigma_\theta$ = 26.5 kg m$^{-3}$
(150 m depth) and were separated from the deep OMZ by an intermediate DO maximum
located at about $\sigma_\theta$ = 26.7 kg m$^{-3}$ (between 200 and 300 m), which rules out a simple vertical
displacement of the vertical gradient.
We analyzed the distribution of zonal and meridional velocity at the depth of the DO minimum
using an eddy identification algorithm as described in *section 3.3.1*. Strikingly, 66% (10 of 15)
of the low-DO events could be related to HBVs, where radii were identified between 20 and 45
km (average 34 km) (Table 1 Fig. 6b and 6d). The radii are substantially smaller than the typical
mesoscale (first baroclinic Rossby radius of deformation) at these latitudes being at the order
of 100 km or more. Instead, these eddies have a confined baroclinic structure, which is
associated to higher baroclinic modes and corresponding smaller Rossby radii of deformation
as is shown in detail in *section 4.4*. The HBVs' horizontal core positions are estimated from the
current velocities and closely match the meridional position of the low-oxygen extremes (cf. 3$^{rd}$



and 6th column for bold marked events in Table 1; Fig. 6a and 6b). Note, that the derived HBVs'
zonal core position range between 23.3°W and 22.9°W, whereas for the low-oxygen extremes,
only the meridional position along the 23°W section can be identified. Notable is the
simultaneous occurrence of two HBVs observed during one cruise in 2009 at positions 10.3°N,
23.2°W and 11.6°N, 22.9°W (Fig. 6a to 6f). These two HBVs were meridionally cut through
their eastern and western flank, respectively, and were both observed with DO concentrations
well below 50 µmol kg$^{-1}$ (Fig. 6a and 6b).

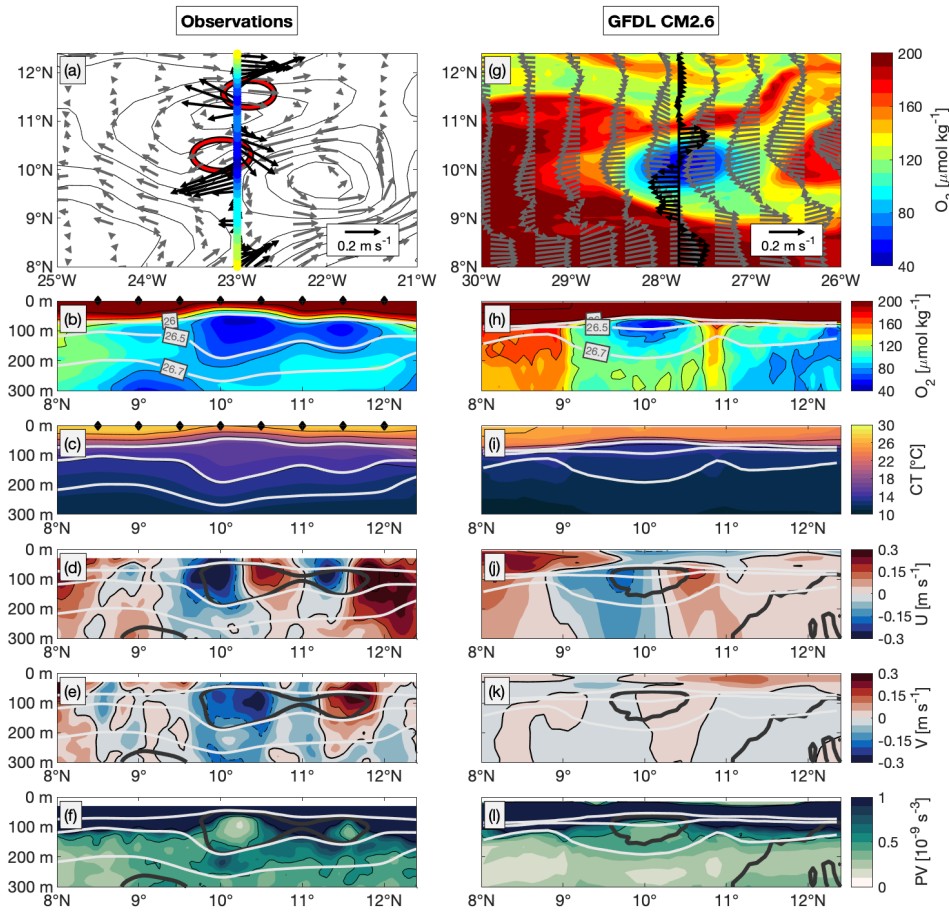


Figure 6: (a) Current velocity (black arrows) and DO (colored dots) at 80 m depth along 23°W
and between 8° and 12°N obtained from along-track shipboard ADCP observations and
CTD-O observations between 23-Jul-2009 and 25-Jul-2009 (cruise Ron Brown 2009, see
Table 1). Grey arrows show geostrophic velocities and black contours show sea level
anomalies from satellite altimetry data on 24-Jul-2009. Red circles denote positions and extent
of the two eddies, identified and reconstructed from shipboard ADCP observations at 80 m.
Latitude-depth sections of (b) DO, (c) conservative temperature, (d) zonal velocity, (e)
meridional velocity and (f) PV between 8° and 12°N obtained from CTD-O observations along
23°W (same period to (a)). Black diamonds at the top of panels (b) and (c) denote actual





latitudes of CTD-O profiles. Thin gray lines in panels (b) to (f) denote surfaces of potential
density. In panels (d)-(e), solid black and dashed black lines denote 0.15 m s$^{-1}$ velocity
intervals. Thick dark gray lines in panels (d)-(f) denote DO contours of 70 µmol kg$^{-1}$. (g)-(l) are
analog to (a)-(f), but taken from GFDL CM2.6 model simulation for model date 23-Mar-0197.
Gray arrows in (g) denote surface velocity. Black arrows denote current velocity and colored
contours show DO distribution both at 77 m depth along ~28°W. (h) to (l) show respective
latitude-depth sections along ~28°W for the same model date. Thick dark gray lines in panels
(j)-(l) denote DO contours of 90 µmol kg$^{-1}$.
Both HBVs were identified to be anticyclonic and subsurface intensified, as shown by the
anomalously weak stratification along 23°W at subsurface depth, which is indicated by the
thickening of isothermal and isopycnal layers at the depth range of the DO minimum core (Fig.
6c). The vertical extent of the HBVs (characterized by displaced isopycnal surfaces or zonal
velocity) reached at least down to about 250 m and covered the vertical extent of the low-DO
cores. The estimated radii are 36 and 31 km and thus considerably smaller than the first
baroclinic Rossby radius of around 90 km at these latitudes. For none of the 10 anticyclonic
HBVs, we could identify any anticyclonic signature from satellite altimetry observations (Fig.
6a). One reason might be that that the resolution of gridded SLA from conventional altimetry
is not sufficient to resolve such small scale features. Another reason could be the fact that the
eddies are strongly confined to the thermocline (below 30 – 50 m) and often do not have a
surface signature.

### 589 4.4 Vertical structure of low-oxygen high-baroclinic mode vorticies

The decomposition of a disturbed density profile into vertical baroclinic modes gives evidence
about both the theoretical radius (Rossby radius) and propagation speed for this disturbed
state (*see section 3.1*). Here, we did a vertical baroclinic mode analysis for the meridional
section along 23°W between 6 and 12°N, which allowed us a direct comparison against the
spatially resolved low-DO HBVs observed during the respective ship sections. The vertical
structure of the first 20 baroclinic modes was obtained from the climatological hydrographic
distribution (Fig. 7a to 7d). For all individual CTD-O profiles, we derived displacement profiles,
$\xi$, and calculated vertical mode amplitudes $x_n$ via modal decomposition (as described in
*section 3.1*). We then clustered all $x_n$ (i) related to a low-DO event (Table 1) and (ii) not related
to a low-DO event (i.e. all other profiles along 23°W between 6 and 12°N), and calculated an
average amplitude distribution (Figure 7e). For low-DO events, we found substantially
enhanced amplitudes at all modes, but particulary at mode 2, 4, 7, 9 and 10 compared to the
average amplitude distribution that is related to no low-DO events (Fig. 7e). These higher
baroclinic modes n = 4 to 10 (exemplarily shown for 9°N, 23°W in Fig. 7a and 7c) for vertical
displacement and pressure/horizontal velocity, respectively) have zero crossings in the upper
few hundred meters, that are of similar vertical length scale compared to the vertical extent of





low-DO SCVs (near-surface to 250 m, see *section 4.2.1*). The lower baroclinic modes (e.g.
mode 2) have a much larger vertical length scale and are not capable of describing the vertical
structure that is related to low-DO HBVs. The corresponding Rossby radius of deformation for
vertical baroclinic modes 4 to 10 was found from $R_{d,4}$ = 47 km to $R_{d,10}$ = 18 km at 6°N, 23°W
and from $R_{d,4}$ = 24 km to $R_{d,10}$ = 9 km at 12°N, 23°W (Fig. 7g). These radii are well below the
first baroclinic Rossby radius of deformation ($R_{d,1}$ = 152 km at 6°N, 23°W and $R_{d,1}$ = 80 km at
12°N, 23°W) and are close to the average radius of 34 km that was identified for the observed
low-DO eddies (cf. Table 1 and *section 4.3*).

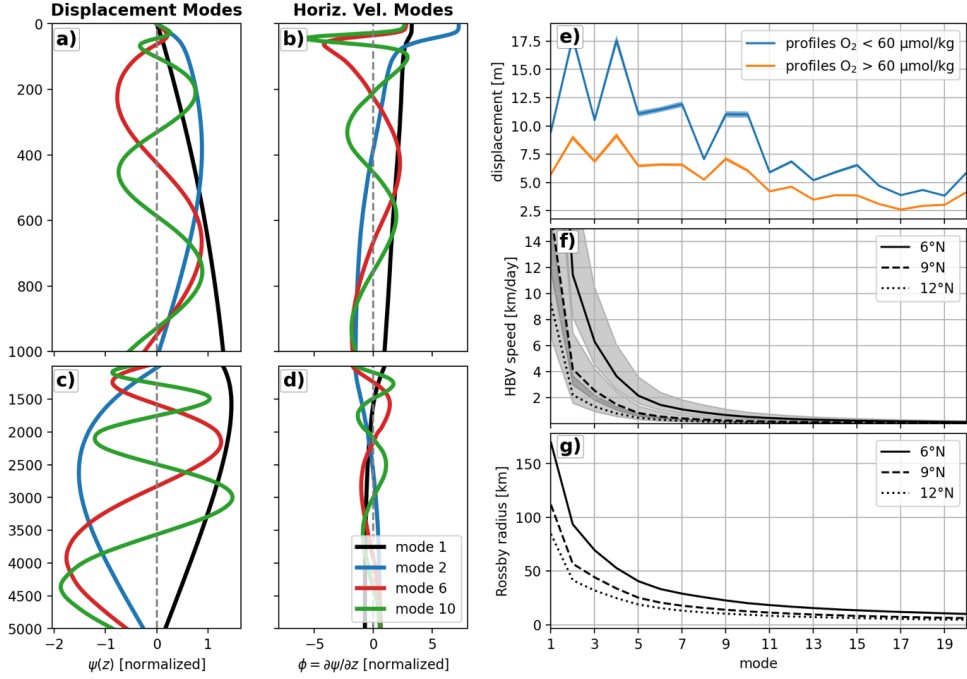


Figure 7: Dimensionless vertical structure functions of baroclinic modes 1, 2, 6 and 10 for (a,
b) isopycnal displacement, $\Psi_n$, and (c, d) horizontal velocity, $\phi_n$, obtained from the
hydrographic profile of the World Ocean Atlas at 9°N, 23°W. (a) and (c) ((b) and (d)) show
depth range 0 to 1000 m (1000 m to bottom). (e) Mean amplitudes of first 20 vertical
displacement modes calculated through modal projection of hydrographic profiles from 23°W
ship sections. Blue solid line denotes mean amplitude distribution, that is related to all
hydrographic profiles with a minimum DO smaller than 60 μmol kg$^{-1}$ in the upper 200 m (i.e.
low-DO events that are summarized in Table 1). Orange solid line denotes mean amplitude
distribution for all other hydrographic profiles along 23°W. Respective shadings denote
standard error of the mean amplitude over all 1000 realizations. (f) Theoretical translation
speed of high-baroclinic Rossby waves (HBVs) for the first 20 vertical modes between 6°N and
12°N along 23°W *(see equation 10)*. The solid black line represents Ro = 0.5, while the shaded



area        indicates        the        range        0.3        <        Ro        <        0.7.
(g) Rossby radii of deformation for the first 20 vertical modes. In both (f) and (g), solid, dashed,
and dotted lines correspond to values at 6°N, 9°N, and 12°N along 23°W, respectively.

### 630    4.5 Source waters of high-baroclonic mode vorticies

The determination of the physical origin of subsurface HBVs, that are associated with the
observed low-DO events, is not straight forward. A backtracking algorithm based on satellite
altimetry observations as used in other studies for more poleward eddies (Chelton et al., 2011;
Schütte et al., 2016a) is not applicable here, since these near-equatorial HBVs are hardly
captured in the respective SLA products (Fig. 6). Instead, we derived water mass
characteristics from all CTD-O profiles (Fig. 4a) located in the two boxes [24°-21°W, 6°-12°N]
and [21°-18°W, 6°-12°N] for a conservative temperature range that corresponded to the depth
range of the shallow DO minimum. A mean profile of absolute salinity was calculated for the
two boxes and was used as a reference in order to calculate anomalies of absolute salinity as
a function of potential density for every single CTD-O profile (Fig. 8). For both boxes, we
clustered the salinity anomaly profiles into two classes, that were defined by the minimum DO
concentration in the upper 200 m to be either below or above the threshold of 60 µmol kg$^{-1}$.

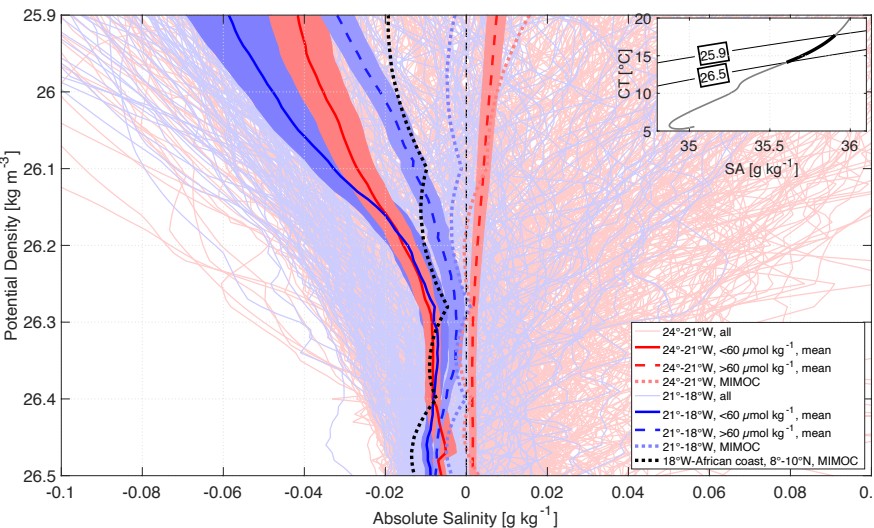


Figure 8: (Large panel) Anomalies of absolute salinity as a function of potential density in the
eastern tropical North Atlantic for two different box regimes (Red: 24°-21°W, 6°-12°N / Blue:
21°-18°W, 6°-12°N). The boxes are highlighted in Fig. 4. The anomalies are referenced to the
mean profile of absolute salinity that was calculated from all hydrographic profiles found in both
boxes. Thin solid lines denote all individual profiles and thick solid (dashed) lines show the
average of the profiles, that are related to minimum DO concentrations below (above)
60 µmol kg$^{-1}$ in the upper 200m. Shadings to the average profiles illustrate the respective





standard errors (see text for details). Blue and red dotted lines denote climatological profiles
for the two boxes. Black dotted line shows the climatological profile for a third box (18°W-
African coast, 8°-10°N), which defines the near-coastal regime off West-Africa. (Inlet panel)
Mean characteristics of absolute salinity versus conservative temperature for the box 24°-
18°W, 6°-12°N, taken from all CTD-O observations in this regime. Thick black line denotes the
characteristics in the potential density range 25.9 to 26.5 kg m⁻³ and is the reference profile for
the anomalies shown in the large panel.
Along-isopycnal gradients of mean salinity are weak (i.e. small spiciness) in the considered
region [24°W-African coast, 6°-12°N], as shown by the water mass characteristics obtained
from the climatological distribution (MIMOC) for the two boxes as well as for the near-coastal
area east of them. The westward salinity increase along isopycnal surfaces is roughly 0.01 to
0.02 g kg⁻¹ per 5° (from the African coast at 17°W to about 22°W) in the potential density range
between 26.1 and 26.4 kg m⁻³. This weak isopycnal gradient does not allow for a differentiation
of water mass characteristics from individual CTD-O profiles.
However, water mass characteristics for low-DO and high-DO profiles were found to be
significantly different from each other, when isopycnally averaging over all respective profiles.
For the western box [24°-21°W, 6°-12°N], low-DO profiles were on average lower in salinity
(compared to high-DO profiles) and they were found to be close to the average salinity anomaly
profile from the eastern box [21°-18°W, 6°-12°N], suggesting that water masses related to low-
DO profiles have their origin closer to the eastern boundary. However, the tropical low-DO
extremes appear in the open ocean far away from the eastern boundary. The westward
intensification of these events (*section 4.1*, Fig. 4), that are often related to HBVs (*section 4.2*),
suggests an unexpected long isolation of the DO depleted water masses in the otherwise
oxygen rich open ocean.
**4.6 Origin & temporal evolution of high-baroclinic mode vorticies based on model**
**simulations**
Outputs from the GFDL CM2.6 ocean model is used to investigate the origin and temporal
evolution of these unusual vorticies. We used the last 20 years of simulations for a regime
similar to that considered in the shipboard observations. From Fig. 2, we already know that the
model captures the main features of the mean state of the oxygen distribution. To assess
whether low-oxygen events occur with similar frequency in the model and whether they are
likewise associated with HBVs, we conducted analyses analogous to those performed on the
observations (*Section 4.1*, Fig. 4; and *Section 4.3*, Fig. 6) using the model data.
First the horizontal DO distribution was calculated by taking the temporal and vertical (0-200 m)
minimum of the simulated DO similar to the observations (Fig. 4b). In the latitude range 6°-
12°N, lowest DO below 30 µmol kg⁻¹ is found close to the African coast (east of 18°W). In



general, the basin wide gradient of minimum DO is positive towards west, being in agreement
with the zonal gradient of the mean simulated DO distribution (Fig. 2b). Strikingly, minimum
DO is lower in the region 30°-24°W, 8°-12°N than in the region east of it (24°-21°W). The
threshold for the DO 10-percentile (100 µmol kg$^{-1}$) does not change over this longitude range,
whereas the mean DO distribution is increasing towards west (lower whiskers versus box
centers in Fig. 4d). The open ocean minimum of the DO distribution that is found in the region
30°-24°W, 8°-12°N is in good qualitative agreement (though located further west) with the
observed DO distribution (Fig. 4b versus Fig. 4a). In the longitude range 24°-21°W, low-oxygen
events are less likely.
From the GFDL CM2.6 model, we identified a HBV with a low-DO core in the near-equatorial
open ocean as exemplarily shown at the position 28°W, 10°N (Fig. 6h to 6n). The spatial extent
is comparable to our observational results (Fig. 6a to 6g). A meridional cross section through
the simulated HBV reveals the low-DO core at 80 m depth (isopycnal surface 26.5 kg m$^{-3}$) with
a lateral extent of about 1° in latitude and a vertical extent between about 50 and 150 m (Fig.
6i). The minimum DO is lower than 60 µmol kg$^{-1}$, whereas DO outside the HBV is at values
above 150 µmol kg$^{-1}$. Distributions of conservative temperature and potential density show
shallowing and deepening isopycnal surfaces above and below the DO minimum, respectively,
indicating a weakened stratification and consequently low PV at the low-DO core (Fig. 6j). The
HBV's velocity signature is strongly confined to subsurface depths and vanishes above 50 m
(Fig. 6k). In particular, surface velocity does not show any coherence with the subsurface
velocity field at depth of the HBV (Fig. 6h). This substantiates our observational results that
these HBVs can hardly be identified from the surface geostrophic velocity field obtained from
satellite observations. The HBV core exhibits low PV water, where minimum PV is found
slightly deeper than the DO minimum (Fig. 6l). This low PV water core is laterally isolated from
the surrounding high PV water, but also separated from the deeper low PV water through an
intermediate PV maximum along the isopycnal surface 26.7 kg m$^{-3}$. This isolation is the
prerequisite for a persistent eddy with a long-life time.
In the following, we present the temporal evolution of the HBV from the time of formation to the
decay. Fig. 9 shows model snapshots with horizontal maps of PV, relative vorticity normalized
by f (so that its magnitude is equal to Rossby number), DO and salinity for four different time
points throughout the HBV's lifetime. Fig. 10 shows time series of different physical and
biogeochemical variables for the HBV core position.



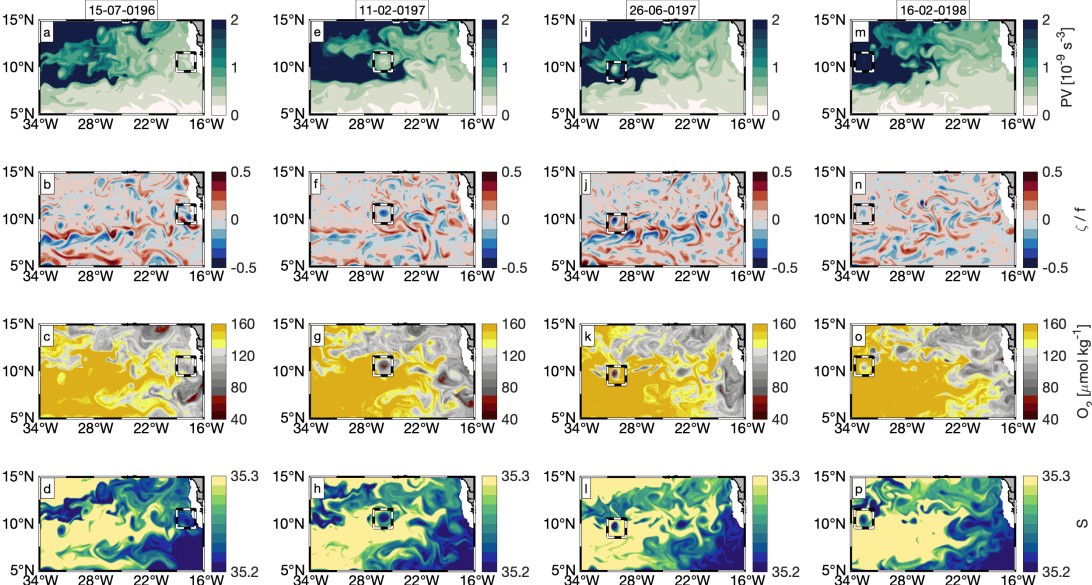

Figure 9: Model snapshots of PV on isopycnal surface 26.6 kg m 3 (top row), relative vorticity
over f, DO and salinity on isopycnal surface 26.5 kg m 3 (second, third and fourth row) for
different phases (different columns) of an anticyclonic HBV (respective time indicated above
each column with T = 0 / 211 / 346 / 580 days: formation / strongest peculiarity / weakening /
decay.  Black-white dashed box in each sub panel denotes HBV position.
The HBV has its origin at the eastern boundary at 18°W, 10°N, where low PV water (Fig. 9a)
with anticyclonic vorticity (Fig. 9b) is deflected offshore and provides the precondition for the
eddy formation. The offshore deflected water carries typical water mass characteristics from
the eastern boundary: low DO and low salinity (Fig. 9c and 9d). During westward propagation
into the open ocean, the HBV enters high PV waters. 211 days after formation, it reaches
26°W, 10.5°N with low PV (Fig. 9e) and high negative relative vorticity (Fig. 9f) in its core. The
coherent eddy is strongly isolated from surrounding high PV water as shown by the intensified
DO minimum (Fig. 9g) and low salinity (Fig. 9h) in its core. In the following 5 months the HBV
propagates further westward, but is disturbed by high PV water, that is advected from the
western tropical Atlantic. This leads to a weakening of the HBV with a smaller low PV core
(Fig. 9i), but still carrying pronounced negative relative vorticity (Fig. 9j), low DO (Fig. 9k) and
low salinity (Fig. 9l) compared to surrounding water. The HBV eventually loses its energy and
decays about 580 days after formation at 33°W, 10°N (Fig. 9m and 9n), where the core water
still appears with anomalous low DO and salinity (Fig. 9o and 9p).





The quick offshore deflection of coastal water, that is associated with the HBV's formation, is
illustrated by the strong change in longitude (Fig. 10a) and by the high propagation speed (Fig.
10d) during the first 50 days. This deflection is more like a pulse rather than an offshore
transport of enclosed water $((U/c)^{-1} > 1$, Fig. 10f), where the HBV stabilizes after that time at
a radius of 50 km (Fig. 10b).

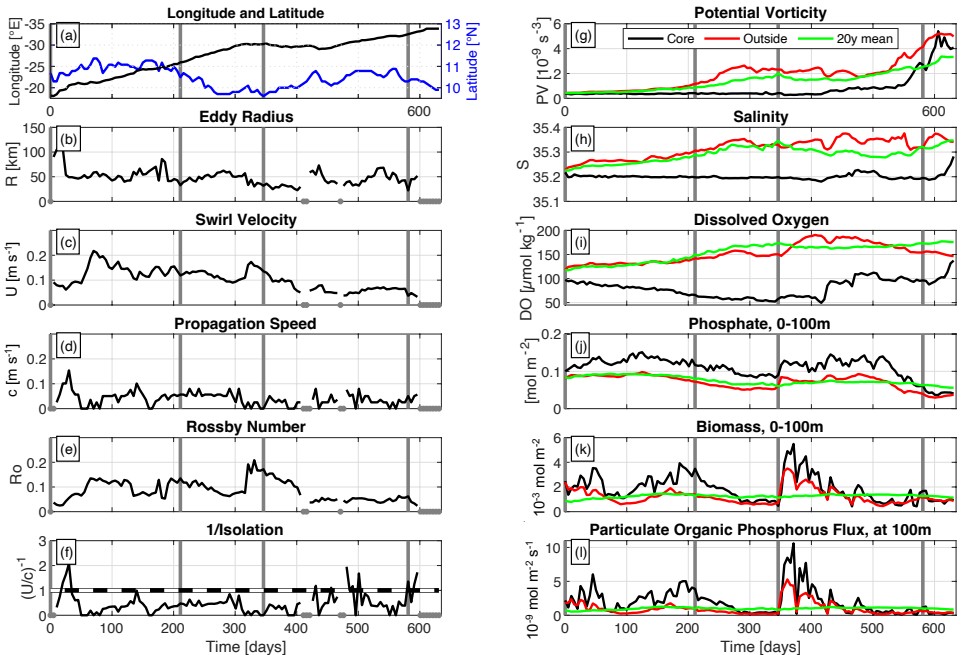


Figure 10: Time series of different variables related to the core of the modelled subsurface
intensified eddy shown in Figs. 4h-4n and Fig. 8. Time is given as elapsed days since eddy
detachment from the African coast. Vertical gray lines in each panel denote time points for
horizontal maps shown in Fig. 8 (0, 211, 346 and 581 days). (a) longitude (black line) and
latitude (blue line), (b) Eddy radius, (c) Eddy swirl velocity, (d) Eddy propagation speed, (e)
Rossby number, (f) Inverse of isolation parameter (black line). Black-white dashed line denotes
threshold, below which the water is trapped in the eddy core (swirl velocity > propagation
speed). (g) Potential vorticity, (h) salinity, (i) DO, (j) phosphate, (k) biomass, (l) flux of
particulate organic phosphorus. Variables are given for the following layers. In panels (a)-(f)
and (h)-(i): isopycnal surface 26.5 kg m⁻³. In panel (g): isopycnal surface 26.6 kg m⁻³. In panels
(j)-(k): integral over 0-100 m. (l) at 100 m. For the right column (panels (g)-(l)), black lines show
value in eddy core, red lines show mean values outside the eddy (average between 1° and 3°
of longitude/latitude around the eddy core position) and green lines show 20-year model mean
that is given at the respective position of the eddy core. In panels (b)-(f), gray dots at zero line
denote time points, where no estimate was possible.

From day 80 to day 300, the HBV continuously propagates westward until 30°W with only slight
changes in latitude (10°-11°N), at a propagation speed of 0.7 m s⁻¹, a swirl velocity between
0.1 and 0.2 m s⁻¹ and a radius of 50 km (Rossby number between 0.1 and 0.2) (Fig. 10a-10f).
The strong isolation $((U/c)^{-1} < 1)$ over that time keeps the core water constantly low in PV



and salinity, while surrounding waters increase in PV and salinity during eddy westward
propagation (Fig. 10g-10h). DO continuously decreases from 97 µmol kg$^{-1}$ to 54 µmol kg$^{-1}$ over
300 days which yields an average DO consumption of 0.14 µmol kg$^{-1}$ d$^{-1}$ (Fig. 10i). In the upper
part of the eddy, enhanced nutrient concentration is associated with increased biomass
production, which leads to enhanced export of organic matter between days 100 and 300 (Fig.
10j-10l). The associated increased respiration and the strong isolation both lead to the
development of this substantial DO deficient zone. The high PV water, that is advected from
the west, acts as a barrier for the HBV and westward propagation stops after day 300 (Fig.
10a and 10d). The HBV is deformed by the high PV water, which likely leads to enhanced
isopycnal and diapycnal mixing at the eddy periphery. In fact, the HBV shrinks between days
300 and 400 as illustrated by the continuously decreasing radius from 50 km to 30 km (Fig.
10b). Though, the core still shows source water characteristics with unaltered low PV and low
salinity, and still holds the DO deficient zone. After day 400, the SCV starts to interact with
surrounding water - partly being low in PV as well - which weakens the isolation of the HBV
core ($(U/c)^{-1} \approx 1$, Fig. 10f) and leads to continuous increase of PV and DO. PV strongly
increases after day 550 and reaches the PV threshold of surrounding water at about day 600,
where the core starts to dissolve as illustrated by the strong increase of salinity and DO after
day 600.

## 5   Discussion

Moored time series of dissolved oxygen (DO) in the near-equatorial Atlantic (4°N up to 12°N)
occasionally show pronounced dips in oxygen concentrations falling significantly beneath the
climatological mean, well below the mixed layer and lasting for several weeks. In addition, we
found that about 8% of all observed CTD-O profiles in the near-equatorial ETNA (25°-15°W,
6°-12°N) appear with anomalous low-DO (< 60 µmol kg$^{-1}$) in the upper 200 m, which is as well
below the climatological DO concentration. Until now, the causes of these extreme low-DO
events have remained unclear. Mesoscale eddies with low oxygen cores - known to occur
farther north around 20°N - are not expected to drive such extensive oxygen-deficient zones
in the near-equatorial region, as they are not believed to persist here as coherent vortices with
lifespans of several months or longer (Chaigneau et al., 2009; Keppler et al., 2018). However,
the majority of these low-DO events (60%) are clearly associated with high-baroclinic
subsurface-intensified eddies (Table 1, Fig. 4, Fig. 5). For the remaining 40%, the velocity and
density distributions did not reveal clear eddy signatures, nor did satellite data - consistent with
all identified vortices, which generally lack a distinct surface signature. However, a connection
to subsurface-intensified eddies cannot be ruled out a priori for these cases.
This underlines that in understanding the Earth system, a better understanding of small-scale



ocean dynamics (smaller than the first baroclinic Rossby radius of deformation) is essential,
as they play a crucial role in the distribution of energy and tracers as well as the regulation of
biogeochemical processes. In particular, below the surface layer - where satellite observations
are ineffective - our understanding of the frequency, magnitude, and impact of these small-
scale ocean dynamics remains limited.
In the vicinity of the equator (< 5°N/S), mesoscale dynamics dominantly appear as horizontally
anisotropic waves (e.g. tropical instability waves) rather than closed circular structures. These
wave-like structures, however, are not isolated enough to effectively transport or develop low-
oxygen environments. The eddies with DO anomalies that we observed are relatively small
and long-lived high-baroclinic vorticies (HBVs). Ship sections along 23°W exclusively revealed
anticyclonic HBVs, whereas both anticyclonic and cyclonic HBVs were found from moored
observations at 11°N, 21°W and in the model.
**5.1 Vertical and horizontal structure of the low-oxygen events and the associated**
**high-baroclinic mode vorticies**
The observed anticyclonic HBVs had a pronounced low-DO core that vertically extended from
the base of the mixed layer down to several hundred meter depth (with minimum DO at depths
between 45 and 90 m). The anomalous horizontal velocity of the observed anticyclonic HBVs
was at maximum (maximum EKE) at the depth of the DO minimum and extended from 50 to
roughly 250 m. Stratification in the observed anticyclonic HBVs' core was weak over this depth
range with upward and downward displaced isopycnals above and below the depth of EKE
maximum, respectively. We found an average radius of about 34 km (between 20 and 45 km)
for the observed HBVs. A decomposition into vertical baroclinic modes showed, that modes 4
to 10 fit best to low-DO events that are related to these HBVs. The associated $4^{th}$ to $10^{th}$
baroclinic Rossby radii of deformation are between 34 and 13 km (at 9°N) and in good
agreement with the observed eddy radii. The observed radii appear well below the first
baroclinic Rossby radius of deformation (more than 100 km in the region) and corresponding
eddies can be considered as higher baroclinic mode vorticies. Rossby numbers were below 1
(about 0.3 – 0.7 found from shipboard observations; 0.4 found in the GFDL CM2.6 model
simulation).
The observed cyclonic HBVs appeared with a stratification maximum at about 150 m and a
cyclonic velocity structure with maximum EKE at a similar depth. Shallow DO minima were
found at 100 and 200 m throughout the transition, but without any clear separation from the
deep OMZ at 300 m. This less intensified DO minimum at 100 m and the missing intermediate
DO maximum at 200 m is a substantial difference to the DO distribution observed within





anticyclonic HBVs. However, enhanced DO consumption has been shown to be a reasonable
driver for DO depletion in a cyclonic HBV, that was observed in the western subtropical North
Atlantic (Li et al., 2008). Pure upwelling or upward mixing of low-DO water from the deep OMZ
cannot explain such vertically homogeneous distribution of low-DO between 100 and 300 m in
cyclonic HBVs. These processes would imply either a shallowing of isopycnal surfaces or a
weakened stratification within this depth range, which is contradictory to the observed
deepening of isopycnals above 150 m, shallowing of isopycnals below 150 m and
consequently the intensified stratification at 150 m (Fig. 5, Event #03). Due to the increased
stratification, the thickness of the intermediate DO maximum layer (that is associated with
isopycnal 26.6 kg m$^{-3}$) is reduced and very likely not resolved by the sparsely distributed
number of DO sensors.
We could not collocate any clear signals in SLA or SST from satellite observations with the in
situ observed HBVs. Shipboard observations showed a strongly weakening velocity signature
toward the surface. Also the simulated HBVs from GFDL CM2.6 model showed similar
characteristics as the observed vorticies and no signature could be found from the surface
velocity. Moreover, the resolution and interpolation scheme for the gridded SLA data likely do
not allow to properly capture geostrophic structures at scales of smaller than about 40 km.
These are likely reasons, why near-equatorial subsurface eddies are hardly identified from
satellite products. If higher resolution satellite products from SWOT will allow to detect the
HBVs remains to be seen, though what we see from the in-situ observed structure and the
model results we conjecture that the high baroclinic mode HBVs tend to "hide" below the
surface/mixed layer base.
**5.2 Origin, lifetime & evolution of the oxygen content of high-baroclinic mode**
**verticies**
Water mass characteristics derived from shipboard observations showed that open ocean
water masses with DO below 60 µmol kg$^{-1}$ in the upper 200 m (often associated with HBVs)
likely originate from the eastern boundary, where SACW contributions exceed those of NACW.
Model results are in agreement as they show the formation of a low-DO HBVs with low PV in
its core off the African coast at about 10°N, 18°W. Hence, it is expected that the generation
mechanism is consistent with previous studies on HBV formation, in which the interaction
between the mean flow and sharp topographic curvature leads to the formation low-PV waters
within the bottom boundary layer, and the shedding of HBVs (D'Asaro, 1988; Molemaker et
al., 2015; Thomsen et al., 2016; Srinivasen et al., 2017; Dilmahamod et al., 2022).
The simulated HBV analyzed here propagated westward far into the open ocean over a



distance of 1,600 km (10°N, 33°W) and lasted for 600 days (average propagation speed of
2.6 km day$^{-1}$). For the observed HBVs, we could not derive propagation speeds in a similar
way. Instead, we followed an approach by Nof (1981) and Rubino et al. (2009), who formulated
the westward translation of isolated high baroclinic eddies on a plane, which is given as a
function of the $n$-th baroclinic Rossby radius of deformation and the Rossby number:

$$C_n = -\frac{1}{3}\beta R_{d,n}^2 (1 - Ro)^{-1} \tag{2}$$

with $\beta$ being the meridional derivative of the Coriolis parameter. Considering a Rossby radius
between 35 and 50 km and a Rossby number between 0.3 and 0.7 (taken as characteristic
scales from the observed HBVs corresponding to the vertical baroclinic mode 4) yields a
propagation speed between 1.1 and 5.4 km day$^{-1}$. This is in good agreement with the
propagation speed obtained from the simulated HBV. Considering the origin of the observed
low-DO HBVs at the eastern boundary (cf. *section 4.4*), a propagation speed at 1.1-5.4
km day$^{-1}$ yields a propagation time of 100 to 500 days to propagate a distance toward 23°W
(around 550 km). The fact, that these high baroclinic low-DO HBVs are not captured by satellite
products, prevents both backtracking to their origin and estimating their lifetime directly.
However, assuming that the HBVs origin close to the African coast and propagate westward
at 1.8 to 4.9 km day$^{-1}$, they would require approximatley 110 to 300 days to travel the 550 km
distance to 23°W.
In contrast to anticyclonic HBVs, cyclonic HBVs only twice in the mooring time series and were
not found in any of the numerous ship sections along 23°W. We may only speculate, that
cyclonic HBVs do not frequently propagate across 23°W due to a much more reduced eddy
life time. They transport anomalously high PV water in their core compared to surrounding
water masses. During westward propagation, the isolation of the core is expected to be
reduced due to the westward increasing PV background gradient in the tropical Atlantic. As
anticyclonic HBVs propagate westward, their low-PV cores are reinforced and remain isolated
from surrounding waters, promoting their longevity. However, encounters with high-PV water
from the western basin may destabilize them, while interactions with other low-PV anticyclones
can enhance their stability. Throughout the long lifetime of the simulated anticyclonic HBVs,
enhanced respiration in the eddy core leads to a strong DO decrease from 92 to 46 µmol kg$^{-1}$
over a period of 260 days, yielding a DO consumption of 0.18 µmol kg$^{-1}$ d$^{-1}$. This fits to our
observational results, where lowest absolute DO concentrations occurred in the open ocean
(24°-21°W) rather than in the region that is located closer to the eastern boundary (21°-18°W).
The here found DO consumption rates are also in good agreement with consumption rates
estimated from observed subsurface intensified anticyclonic eddies (0.19 ± 0.08 µmol kg$^{-1}$ d$^{-1}$)



that originate from the Mauritanian upwelling system and propagate westward at about 18°N
(Schütte et al., 2016). We shall note, that DO was observed close to anoxic conditions on the
shelf of Senegal at about 14°N at depths of about 20 m (Machu, 2019). However, these water
masses are at much shallower depth and lighter densities and are very likely not the source
region for the low-DO core of the here described offshore HBVs. However, the here described
low-DO eddies, characterized with low PV waters in their cores, likely have their origin at the
eastern boundary with the bottom boundary layer identified as the source of this low PV waters.

## 6 Summary and conclusion

We shall summarize the following take home messages to the reader:

(i) Distribution and occurrence of low-DO events:

In the near-equatorial North Atlantic (25°-15°W, 6°-12°N), about 8% of all CTD-O profiles occur with a DO concentration of less than 60 μmol kg$^{-1}$ in the upper 200 m, which is well below the climatological DO concentration. These extreme low-DO events are more frequent and more intensified in the open ocean (30°-21°W) compared to the region east of it (21°- coast of West Afrcia). Unprecedented low-DO concentrations were found with 1 μmol kg$^{-1}$ at 80 m depth at the mooring located at 11°N 23W as well as 17 μmol kg$^{-1}$ (8°N, 23°W) and 29 μmol kg$^{-1}$ (9°N, 21°W) observed with ship based measuremnts.

(ii) Low-DO events are related to subsurface intensified submesoscale coherent vortices:

We found 66% of open ocean low-DO events to be related to subsurface intensified submesoscale coherent vortices, where anticyclonic rotation appeared as the dominant eddy type. These vortices have a high baroclinic vertical structure, associated to vertical baroclinic modes 4 to 10, and are confined to the upper 250 m. In situ velocity observations revealed an average radius of 34 km, which is well below the first baroclinic Rossby radius of deformation (O(100 km)), but agrees well with Rossby radii of the higher baroclinic modes 4 to 10 (34 to 13 km at 9°N). Despite the small length scales, the Rossby number of the vortices is below 1, assigning them to the dynamical range of mesoscale variability.

(iii) Origin and life time:

The vortices most likely originate from the eastern boundary. They can propagate far into the open ocean with a propagation speed of 1.8 - 4.9 km day$^{-1}$, reaching a life time of more than half a year (it took around 100 to 500 days to propagate the 550km distance towards 23°W). This is much longer than currently suggested considering the highly dynamical area and the proximity to the equator. Model simulations even show a life time of up to 1.5 years. Cyclonic eddies with low-



oxygen cores were less frequent than anticyclonic eddies. Cyclonic eddies were
not found in ship sections along 23°W, but in the minority of all low-DO extreme
events from moored observations at 11°N, 21°W.

(iv) Impact of the vorticies on DO and biogeochemistry:

The unexpected long-life times of the near-equatorial vortices goes along with a
strong isolation of their low-PV core from surrounding water. These vortices are
capable of forming a DO deficient zone in their core, since enhanced primary
production and remineralisation leads to increased DO consumption (0.18 µmol kg$^{-1}$
943           day$^{-1}$ at least for the simulated anticyclonic vortices).

(v)  Detection of near-equatorial vortices with remote sensing satellites:

Near-equatorial vortices are hardly detectable by conventional satellite altimetry
observations, which precludes a backtracking of these eddies.  New observations
are desirable to verify whether the new SWOT mission can capture such SCV,
although a strong surface signal is not expected due to the mainly subsurface
structure (also supported by the model).

Subsurface coherent vortices in the near-equatorial ocean have been so far overlooked in
driving DO deficient zones. The long-lived vortices appear unexpectedly quite regularly given
theoretical considerations and are able to generate hypoxic regimes in the open ocean, which
may impact on pelagic fish, biodiversity and biogeochemical cycles. They are typically not
tracable in satellite products, which makes a collocation of satellite data with in-situ
observations (CTD-O, Argo profiles, moored observations) hardly possible. The comparatively
coarse resolution of satellite observations might instead lead to a wrong collocation of the
subsurface low-DO events with larger surface intensified mesoscale structures nearby. The
mechanisms for the generation of these near-equatorial low-DO eddies remain an open
question. So far, we here identified a potential source region and provided a first insight about
the dynamics (life time, baroclinicity, isolation) of these eddies. A more comprehensive
investigation from high resolution ocean circulation models - coupled to biogeochemistry -
would shed light onto the generation. Further, the study of the temporal evolution of dominant
vertical baroclinic modes throughout the eddies' life cycle would contribute to a better
understanding of the eddy dynamics and stability. Moreover, the interdisciplinary view on
changes in biogeochemical processes would increase the understanding about the impact on
biogeochemistry. The in-situ tracking and observation of these eddies over their life cycle is
challenging, but would provide key information to validate the simulation of these eddies.




**7   Data availability**

The assembled shipboard measurements (27 research cruises) and moored data used in this paper are available and collected at https://doi.pangaea.de/XXXX. The used satellite altimetry data is provided by Marine Copernicus (https://marine.copernicus.eu) can be downloaded at https://doi.org/10.48670/moi-00148. The used gridded climatological hydrography and oxygen from the World Ocean Atlas 2023 (WOA23), is available at NOAA under: https://doi.org/10.25921/va26-hv25. The monthly, isopycnal and mixed-layer ocean climatology (MIMOC) used is available at https://www.pmel.noaa.gov/mimoc/. The Model data will be made openly accessible via the GEOMAR website https://data.geomar.de where data is uniquely identifiable via handle assignment (PID) and will be accessible per download.

**Author contributions**

Conzeptualization: FS, JH, PB, Data curation: JH, IF, FS, Formal analysis and methodology: JH, IF, MS, FS, AB, FD, Funding acquisition: PB, Writing – original draft: JH, FS, IF, Writing – review and editing: FS, JH, IF, AB, FD, MS, PB

**Competing interests**

The contact author has declared that none of the authors has any competing interests.

**Acknowledgements**

We thank the crew and the "Leitstelle Deutsche Forschungsschiffe" for supporting the numerous expeditions in th eastern tropical North Atlantic that have made this work possible. Research cruises with RV Meteor, RV Maria S. Merian were funded by the Deutsche Forschungsgemeinschaft as part of Sonderforschungsbereich 754 "Climate-Biogeochemistry Interactions in the Tropical Ocean" and through other projects such as the EU H2020 TRIATLAS project (grant agreement 817578) funded by the German Federal Ministry of Education and Research (BMBF). Moored velocity and oxygen observations were partly acquired in cooperation with the PIRATA project, and we would like to thank B. Bourlès, R. Lumpkin, C. Schmid, and G. Foltz for their help with mooring work and data sharing. We thank the captains and crew of the RV *Maria S. Merian*, RV *Meteor*, RV *Poseidon*, and RV *L'Atalante* as well as our technical group for their help with the fieldwork.



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





**Table 1**. Low-DO events (below 60 µmol kg$^{-1}$) found in the upper 200 m during meridional CTD-O ship sections along 23°W between 7° and 12°N. Only those low-DO events are listed, where meridional sections of DO, hydrography and velocity were available (spanning a latitude range of minimum 3°). Columns from left to right denote DO minimum between 0 and 200 m, corresponding depth, latitude and research cruise with date of the CTD-O profile. The last three columns denote type, core position and radius of related eddy, that was analyzed with the eddy identification method. ACE events are marked in bold (the abreviation ACME stands for anticyclonic mode water eddy). As an example, the event in the third row (Meteor 119/1, 17-Sep-2015) is presented in Fig. 4.

| DO minimum [µmol kg$^{-1}$] | Depth [m] | Latitude [°N] | Cruise ID (Date) | Eddy type | Eddy core position | Radius [km] |
|---|---|---|---|---|---|---|
| **17** | **59** | **8,0** | **Meteor 116/1 (22-May-2015)** | **ACME** | **8.3 °N 23.1 °W** | **33** |
| 37 | 63 | 11,5 | Meteor 116/1 (21-May-2015) | - | - | - |
| **42** | **71** | **8,0** | **Meteor 119/1 (17-Sep-2015)** | **ACME** | **8.0 °N 23.3 °W** | **38** |
| **44** | **45** | **10,0** | **Ronald H. Brown PNE09 (24-Jul-2009)** | **ACME** | **10.3 °N 23.2 °W** | **36** |
| **47** | **69** | **10,5** | **Polarstern PS88.2 (08-Nov-2014)** | **ACE** | **10.3 °N 23.2 °W** | **37** |
| **48** | **77** | **11,5** | **Ronald H. Brown PNE09 (24-Jul-2009)** | **ACME** | **11.6 °N 22.9 °W** | **31** |
| 52 | 75 | 11,5 | L'Atalante IFM-GEOMAR 4 (11-Mar-2008) | - | - | - |
| 53 | 67 | 11,0 | Meteor 097/1 (30-May-2013) | - | - | - |
| **54** | **93** | **7,0** | **Meteor 068/2 (04-Jul-2006)** | **ACME** | **7.1 °N 23.0 °W** | **20** |
| 55 | 65 | 10,5 | Merian 018/3 (25-Jun-2011) | - | - | - |
| **56** | **74** | **7,0** | **Ronald H. Brown PNE06 (30-Jun-2006)** | **ACME** | **7.0 °N 23.0 °W** | **45** |
| 57 | 71 | 11,5 | Meteor 130/1 (03-Sep-2016) | - | - | - |
| **58** | **82** | **11,0** | **Meteor 105/1 (10-Apr-2014)** | **ACME** | **11.0 °N 23.2 °W** | **60** |
| **58** | **73** | **11,5** | **Merian 022/1 (15-Nov-2012)** | **ACME** | **11.6 °N 23.0 °W** | **37** |
| **58** | **79** | **10,5** | **Meteor 106/1 (24-Apr-2014)** | **ACME** | **10.4 °N 23.2 °W** | **33** |