# Peer review of "Hidden vortices: Near-equatorial low-oxygen extremes"

_EGUsphere, 2025_

## Author Comment (AC1)

*Response to Reviewer #1*

We sincerely thank the reviewer for their careful reading of the manuscript and their thoughtful and constructive comments. We appreciate the recognition of the scientific value of the observational dataset and the phenomenon we refer to as high-baroclinic-mode vortices (HBVs) in the near-equatorial North Atlantic. The reviewer raises important points, especially concerning the interpretation of HBV longevity based on model output and the potential limitations of the biogeochemical model representation. We value this critical perspective and have taken care to address these concerns in detail. In particular, we have clarified the limitations of the model, incorporated additional observation-based diagnostics where feasible, and revised key parts of the manuscript accordingly. Below, we respond point-by-point to each of the reviewer's comments. Reviewer comments are reproduced in **black**, followed by our responses in **green**. Changes to the manuscript are described where appropriate and indicated in *italics*.

**Anonymous Referee #1**

*Note:* The reviewer comments are in **black**, our responses are in **green**

**The authors describe a vertically and horizontally spatially constrained low dissolved oxygen (DO) event observed in the near-equatorial North Atlantic using moored and repeated ship-based observations, and investigate its origin using output from a high-resolution coupled climate model with a simplified Biogeochemical component. The manuscript provides a detailed description of the observational data and analysis methods employed. I find the observational documentation of the phenomenon the authors refer to as HBV to be of sufficient scientific value to merit publication.**

Thank you very much for this assessment.

**One major concern is that the authors argue that the long lifespan is one of the key characteristics of the observed HBVs, yet this conclusion appears to rely solely on the model output. As evidenced in the comparison between the model and observations presented in the manuscript, there may be non-negligible biases in the spatiotemporal variability of DO anomalies (see also my later comments). It should also be noted that general ocean circulation models tend to exhibit reduced dissipation of mesoscale/submesoscale structures, potentially leading to artificially prolonged features. Furthermore, the MiniBLING model employed here does not account for the diverse and complex remineralization processes that drive oxygen consumption in the mesopelagic zone.**

We thank the reviewer for raising this important concern. We would like to emphasize that our conclusions regarding the longevity of HBVs are not solely based on model results. Observational evidence, including the salinity-based analysis shown in Fig. 8, provides independent support for our interpretation, with salinity acting as a conservative tracer that confirms the coastal origin of the eddies and their offshore persistence (and we have also added, as you helpfully suggested, nitrate as an additional observational indicator in a new plot - see below or new figure 8).

The model is used as (or meant to be) an additional tool to complement the observations and to test the plausibility of the proposed mechanisms. We agree that the model, like any ocean circulation model, has limitations and does not fully capture the complexity of remineralization processes. To acknowledge this, we have added a statement in the revised manuscript discussing potential biases:

Line 770: *The model tends to slightly underestimate PV and the associated $O_2$ anomalies, indicating somewhat weaker eddy coherence compared to observations. At the same time, due to reduced dissipation in the circulation model, the lifespan of the eddies is slightly prolonged. Additionally, the MiniBLING model does not fully account for remineralization processes in the mesopelagic zone, which likely leads to an underestimation of oxygen consumption. Taken together, this implies that HBVs in the model appear with weaker anomalies but with an artificially prolonged lifespan, which we consider in our interpretation of the results.*

**To strengthen the argument for the longevity of HBVs based on observations, it would be beneficial to incorporate additional evidence, such as analyses using Apparent Oxygen Utilization (AOU), which carries information related to water mass age. If available, supplementary water mass diagnostics using other tracers (e.g., nitrate, phosphate) would also be valuable in corroborating the persistence of low-DO waters associated with HBVs, at least as circumstantial evidence.**

We thank the reviewer for this excellent suggestion. Following the comment, we have added a new plot (new Figure 8b) showing observed oxygen concentrations, the depth of the oxygen minimum, and the corresponding nitrate profiles from CTD casts taken inside and outside of low-oxygen events. These data reveal substantially lower oxygen between 80 - 250 m, accompanied by elevated nitrate concentrations inside HBVs, consistent with ongoing biological remineralization and hence "older" water. This supports the interpretation that HBVs contain persistent, isolated water masses rather than representing only short-lived anomalies. We believe that this addition strengthens our observational evidence and nicely complements the model-based findings, as now discussed in the end of Section *4.5 Source water of high barocline vortices* as follows:

Line 711 and following: *To further support the persistence and age of HBVs, we analyzed CTD observations of oxygen and nitrate inside and outside of low-oxygen events. Fig. 8b shows the median oxygen profiles for CTD casts with a minimum in the upper 200 m of the water column below 60 μmol/kg (blue curve) and those above 60 μmol/kg (orange curves). In the mixed layer, oxygen concentrations are higher in the blue curve than in the orange curve, likely indicating increased biological productivity over the HBVs. The red stars indicate the depth of the observed oxygen minima clustering between 80 to 120m depth. Corresponding nitrate profiles are shown in turquoise (<60 μmol/kg oxygen) and yellow (>60 μmol/kg oxygen). The results reveal substantially lower oxygen concentrations between 80 - 250 m inside HBVs, accompanied by elevated nitrate levels, consistent with ongoing biological remineralization and hence "older" water. This observational evidence indicates that HBVs consist of persistent, isolated water masses rather than short-lived anomalies.*

[Figure]

New figure 8b caption:

*Figure 8b:* *The blue curve shows the median of all oxygen CTD profiles with a minimum below 60 μmol/kg in the upper 200 m. The red stars indicate depths and dissolved oxygen concentrations of these minima. Orange curves represent profiles with a minimum above 60 μmol/kg. Shaded areas indicate the standard deviation. The turquoise line depicts the mean nitrate profile for the profiles with oxygen minima below 60 μmol/kg, and the yellow line shows the mean nitrate profile for the profiles with minima above 60 μmol/kg.*

**Minor Comments:**

**Does the MIMOC dataset include dissolved oxygen? I could not find oxygen data in the source referenced by the authors (https://www.pmel.noaa.gov/mimoc/). If my understanding is correct, what dataset was used to generate Fig. 2a?**

We thank the reviewer for pointing this out. You are correct - the MIMOC dataset publicly available does not provide dissolved oxygen. This was an oversight in the initial version of the manuscript. All oxygen figures with regard to spatial patterns, including Fig. 2a, were generated using the World Ocean Atlas 2023 (WOA23). We have corrected the data source accordingly and removed all references to MIMOC in the revised manuscript.

**Regarding Fig. 2a and 2b, it would further strengthen the manuscript if the authors could include a comparison map of the depth at which the oxygen minimum occurs.**

We thank the reviewer for this suggestion. This is indeed a useful piece of additional information. Rather than adding another panel to the already large Figure 2, we have incorporated this information into the new Figure 8b. In the figure caption, we added: "*The red stars indicate depths and dissolved oxygen concentrations of these minima.*"
In the manuscript text, we also added the following at Line 721: "*The red stars indicate the depths and dissolved oxygen concentrations of the observed oxygen minima, clustering between 80 and 120 m depth.*"

**Line 335: Is it valid to assume that the barotropic component is zero? In fact, the flow estimated from SLA (e.g., the gray arrows in Fig. 6a) may be significantly influenced by the barotropic component. If the barotropic flow cannot be assumed to be zero, wouldn't it be appropriate to subtract the barotropic velocity (approximated by vertically averaged velocity) from the observed velocity profiles as part of the preprocessing?**

Thank you for pointing this out. Typically, the barotropic component is negligible compared to the baroclinic components in the tropics, and SLA is clearly dominated by the baroclinic signal. However, we cannot estimate the barotropic signal since only velocity data from the upper 1000 m are available. The vertical average over the upper 1000 m is dominated by the first baroclinic mode, which has a zero crossing at around 1500 m depth. We believe that assuming a zero barotropic component is the best approach in this case.

**Line 376: *Castelao and Johns (2011)* and *Castelao et al. (2013)* are not included in the reference list. Please double-check that all cited works are properly listed in the bibliography.**

Thank you. Done.

**Line 385 ("The optimal eddy center allows…"): The meaning of this sentence, particularly the latter part, is unclear. Please revise for clarity.**

Yes, the sentence was not easy to understand. We changed it to: (Line 385) *Identifying the optimal eddy center allows us to analyze the circular (azimuthal) velocity around it. From this, we determine the eddy radius as the distance from the center where this velocity reaches its maximum - effectively separating the inner core of the eddy from its outer ring.*

**Line 407: The method for estimating the propagation speed is not clearly described. Please revise this section to clarify how the speed was calculated.**

We changed it to: (Line 412) *To estimate the propagation speed (c), we tracked the eddy core positions at each time step, defined by the streamfunction minimum. The speed was calculated as the horizontal distance between two successive eddy centers divided by the time interval between them.*

**Figure 3c,d: What does the x-axis represent? Please add labels or clarify in the caption.**

It is the time. We included in the caption: *(date on the x-axis)*

**Line 585: In the discussion of discrepancies between ship-based and satellite-derived observations, spatial resolution is indeed important, but temporal resolution is also critical. Note that the raw satellite data used for gridding does not have daily temporal resolution. In addition, what is the reason for omitting near-surface velocities in Fig. 6d and 6e? If such data are available, do satellite-derived velocities correspond better to near-surface velocities from ship-based observations, or to vertically averaged velocities?**

Thank you very much for this valid and important comment. We agree that, in addition to spatial resolution, the temporal resolution of gridded satellite altimetry also plays a crucial role. The limited temporal resolution (multi-day interpolation) can lead to small-scale, short-lived, or fast-moving mesoscale structures such as HBVs being missed or inadequately resolved. We have now addressed this point more explicitly in the revised manuscript:

*Line 594 One reason might be that that the resolution of gridded SLA from conventional altimetry (in time and space) is not sufficient to resolve such small-scale features.*

Regarding the velocity data shown in Fig. 6d and 6e: these are based on ADCP measurements from hull-mounted instruments. Due to blanking distance and flow distortions around the ship's hull, reliable velocity estimates are typically only available from depths of about 18 m and deeper. Therefore, we did not include near-surface currents in the figure, as they could not be reliably measured in our dataset.

However, a direct comparison to satellite-derived geostrophic velocities is meaningful, since the ADCP measurements at ~18 m still lies within the mixed layer and thus provide a reasonable representation of near-surface currents. We expect the main discrepancies between ship-based and altimeter-derived velocities to arise not from the ADCP sampling depth but from the coarse temporal and spatial resolution of gridded SLA products, which do not adequately resolve small-scale velocity structures such as HBVs - one of the points we want to highlight with this paragraph. Ageostrophic velocities are also a potential source of discrepancies, but likely of secondary importance in the context and resolution of Fig. 6.

**Line 626: Does this refer to Equation 9? Please clarify.**

That was indeed unfortunate, and the wrong equation was referenced in this case. We have now restructured the entire paragraph slightly and corrected the reference to point to the appropriate equation. The revised version starts at line 366.

*For the observed SCVs, we could not derive propagation speeds in a similar way. Instead, we followed an approach by Nof (1981) and Rubino et al. (2009), who formulated the westward translation of isolated high baroclinic eddies on a plane, which is given as a function of the $n$-th baroclinic Rossby radius of deformation and the Rossby number:*

$$C_n = -\frac{1}{3}\beta R_{d,n}^2 (1 - Ro)^{-1} \qquad (9)$$

*with $\beta$ being the meridional derivative of the Coriolis parameter.*

**Line 626 ("The solid black line represents…"): In Fig. 6f, I can only identify the line representing HBV speed. Could you clarify what this sentence is referring to?**

This is correct. We rephrased the sentence and now it is clear:

Line 652: *The solid, dashed and dotted black lines represent Ro = 0.5, while the shaded area indicates the range 0.3 < Ro < 0.7.*

**Line 603: Since Fig. 6 only displays modes 6 and 10, it is not possible to assess the behavior across modes 4 to 10. Please revise this sentence or clarify with additional figures if necessary.**

Thank you very much. That was unclear. We have changed the sentence as follows starting in line 632:

*These higher baroclinic modes n > 4 (exemplarily shown here for mode 6 and 10 for 9°N, 23°W in Fig. 7a and 7c for vertical displacement and pressure/horizontal velocity, respectively) have zero crossings in the upper few hundred meters, that are of similar vertical length scale compared to the vertical extent of low-DO HBVs (near-surface to 250 m, see section 4.2.1).*

**Line 692 (regarding Fig. 4a and 4b): There appears to be a stark difference in the zonal distribution of the strength of low-DO extremes (i.e., the lower end of the dots) between the observations and the model. The observations show the most pronounced low DO events offshore (around 24–21°W), whereas the model indicates such events occur closer to the coast. This inconsistency could point to a potentially significant model bias—possibly arising from the model misrepresentation of HBV origins, trajectories, or associated biogeochemical processes along the path toward 23W. Even though the mean fields (e.g., Fig. 2 and the medians in Fig. 4c) appear consistent, this does not guarantee that the model correctly reproduces the variability targeted in this study (e.g., the dot distribution in Fig. 4c, d). While observational data may be limited and subject to sampling bias—potentially explaining some of the discrepancies—this possibility should be explicitly considered. I encourage the authors to justify the model's suitability for this analysis, for example, by performing a model–observation comparison using pseudo-observations from the model. Alternatively, as noted earlier, the conclusions should not rely solely on the model results and should incorporate more observation-based evidence.**

We thank the reviewer for the careful examination of Fig. 4 a, b. We note that these panels compare individual shipboard profiles with model climatology over 20 years. Because the observations represent snapshots of particular events while the model averages over a longer period, differences in the zonal distribution of low-DO extremes are expected. This does not indicate a systematic model bias, but rather arises from the different sampling approaches.

To clarify this point, we have added text in the manuscript:

Line 748: *It should be noted that Fig. 4a,b compare individual shipboard observations with the 20-year model climatology. Observations represent snapshots of specific events, whereas the model averages over a longer temporal period. Consequently, apparent differences in the*

*zonal distribution of low-DO extremes are expected and do not necessarily indicate a systematic model bias.*

We would like to point out that we have already performed such pseudo-observation comparisons with the model, which largely support our conclusions, while acknowledging that eddies and extreme events cannot occur synchronously in the model and observations. This is illustrated in Figures 6 and 9. Finally, we emphasize that our conclusions are not solely based on the model. Observational data, including the salinity-based analysis in Fig. 8, provide independent support for our findings. In addition, following the reviewer's suggestion, we have now included nitrate as an additional observational indicator, providing further evidence to corroborate the model-based results (see page 1 and 2 of this review).

**Lines 606, 776, 947: Please define "SCV" upon first usage.**

This was an error and should have read HBV. The mentioned instances have been corrected.

**Line 825 ("Rossby numbers were below 1"): Is there a corresponding figure showing the Rossby number distribution? If so, please reference it.**

Thank you for this remark. An exemplarily Rossby number from the model simulation are shown in Fig. 8e. For the shipboard observations, no figure is included in the manuscript, but the Rossby numbers were calculated (values of about 0.3 - 0.7) and are therefore reported in the text. To clarify this, we have revised the text to explicitly state that the observational Rossby numbers are not shown in a figure, whereas the model values are illustrated in Fig. 8e.

Line 825: *Rossby numbers were below 1 (about 0.3 - 0.7 found from shipboard observations, not shown; around 0.4 found in the GFDL CM2.6 model simulation, exemplarily shown in Fig. 8e).*

**Line 859: Please define "SACW" and "NACW" when first mentioned.**

Done

References:

Nof, D. (1981). On the β-Induced Movement of Isolated Baroclinic Eddies. *Journal of Physical Oceanography, 11*(12), 1662-1672. https://doi.org/10.1175/1520-0485(1981)011<1662:OTIMOI>2.0.CO;2

Rubino, A., Dotsenko, S., & Brandt, P. (2009). Nonstationary Westward Translation of Nonlinear Frontal Warm-Core Eddies. *Journal of Physical Oceanography, 39*(6), 1486-1494. https://doi.org/10.1175/2008JPO4089.1

---

## Author Comment (AC2)

*Response to Reviewer #2*

We sincerely thank the reviewer for their thoughtful and constructive comments. We greatly appreciate the positive overall assessment of our manuscript and the recognition of the observational and modeling efforts involved in studying high-baroclinic-mode vortices (HBVs) in the eastern tropical North Atlantic. The reviewer's detailed feedback has helped us clarify and improve the manuscript. We have addressed all suggestions and concerns carefully, and we believe the revised version of the manuscript is substantially improved as a result. Below, we respond point-by-point to each of the reviewer's comments. Reviewer comments are reproduced in **black**, followed by our responses in **green**, and changes to the manuscript are indicated in *italics* and described where appropriate.

**Anonymous Referee #2**

*Note:* The reviewer comments are in **black**, our responses are in **green**

**This manuscript presents a compelling study of high-baroclinic-mode vortices (HBVs) in the eastern tropical North Atlantic, combining shipboard and moored data with eddy-resolving model output. The authors describe how HBVs (whose subsurface cores are isolated from surface turbulent processes) transport low-oxygen water masses offshore from the eastern boundary. The study both provides evidence for the physical advection of low-O₂ water but also considers ongoing oxygen consumption via remineralization along the vortices' trajectory. These dynamics are discussed in the context of their potential implications for biogeography and biogeochemical cycling in the region. The observational challenge of capturing HBVs, given their relatively small spatial scale and intermittent frequency of generation, is well acknowledged. In that light, the dataset compiled and analyzed here is impressive and already provides a valuable contribution to the literature.**

Thank you very much for this respectful word.

**The use of numerical modeling to complement the observations is also appreciated, though I raise a few questions below regarding the model's ability to resolve these features.**

As you already mentioned, the model is used as an additional tool to complement the observations and to test the plausibility of the proposed mechanisms. We agree that the model, like any ocean circulation model, has limitations and does not fully capture the complexity of remineralization processes. To acknowledge this - since this point was raised by the other reviewers - we have added a few statements in the revised manuscript discussing potential biases (these are discussed in more detail further below in this review, e.g., the now changed L124, L330, and L770 in the manuscript).

**Overall, I find the manuscript suitable for publication, pending some minor considerations centered around the following suggestions:**

**The authors might consider supplementing their HBV identification and tracking with additional water mass tracers such as spiciness and apparent oxygen utilization (AOU). These metrics, particularly spiciness (which is conserved along isopycnals), can provide clearer insight into the origins and evolution of the anomalies. For example, a panel showing spiciness in Figure 8 could strengthen the interpretation.**

We thank the reviewer for this valuable suggestion. We analyzed nitrate as an additional tracer (as this was also suggested by another reviewer) and included a new plot (new Figure 8b) showing observed oxygen concentrations, the depth of the oxygen minimum, and the corresponding nitrate profiles from CTD casts taken inside and outside of low-oxygen eddies. These data reveal substantially lower oxygen concentrations between 80 - 250 m, accompanied by elevated nitrate levels inside HBVs, consistent with ongoing biological remineralization and thus "older" water. This supports the interpretation that HBVs represent persistent, isolated water masses rather than short-lived anomalies. We believe that this addition strengthens our observational evidence and nicely complements the model-based findings, as now discussed at the end of Section 4.5 (*Source water of high-barocline vortices*). In addition, we emphasize that our conclusions regarding the longevity of HBVs are not solely based on model results. Observational evidence, including the salinity-based analysis in Fig. 8, provides independent support for our interpretation, with salinity acting as a conservative tracer that confirms the coastal origin of the eddies and their offshore persistence. The figure 8b and a new paragraph are included as follows:

Line 711 and following: *"To further support the persistence and longevity of HBVs, we analyzed CTD observations of oxygen and nitrate inside and outside of low-oxygen events. Fig. 8b shows the median oxygen profiles for CTD casts with a minimum in the upper 200 m of the water column below 60 µmol/kg (blue curve) and those above 60 µmol/kg (orange curves). Mixed layer oxygen concentrations for both cases indicate increased near-surface biological productivity of HBVs compared to outside of HBVs. The red stars indicate the depths of the observed oxygen minima clustering between 80 to 120m depth. Corresponding nitrate profiles are shown in turquoise (<60 µmol/kg oxygen) and yellow (>60 µmol/kg oxygen). The results reveal substantially lower oxygen concentrations between 80 - 250 m inside HBVs, accompanied by elevated nitrate levels, consistent with enhanced accumulated biological remineralization due to enhanced productivity and/or "older" water. This observational evidence indicates that HBVs consist of persistent, isolated water masses rather than short-lived anomalies."*

[Figure]

New caption: ***Figure 8b:*** *The blue curve shows the median of all oxygen CTD profiles with a minimum below 60 μmol/kg in the upper 200 m. The red stars indicate depths and dissolved oxygen concentrations of these minima. Orange curves represent profiles with a minimum above 60 μmol/kg. Shaded areas indicate the standard deviation. The turquoise line depicts the mean nitrate profile for the profiles with oxygen minima below 60 μmol/kg, and the yellow line shows the mean nitrate profile for the profiles with minima above 60 μmol/kg.*

**I have questions around the decision to define HBV events using an arbitrary 10th percentile threshold (e.g., in Figures 3 and 4). This approach may flag low-oxygen "anomalies" even in relatively quiescent regions with little true HBV activity. In Figure 3a, for instance, I find only the event between 2023–2024 particularly convincing. It may be worth considering alternative detection criteria, such as thresholds based on standard deviations or interquartile ranges, which could provide a more statistically grounded definition of outliers.**

Thank you for pointing that out. First, we would like to clarify that the majority of low-oxygen eddies are not detected solely using a percentile threshold. We define a low-DO extreme event in the CTD data as any profile with a minimum DO below 60 μmol kg$^{-1}$ in the upper 200 m (e.g., the data shown in Figure 4). This threshold was chosen because values below 60 μmol kg$^{-1}$ in the upper 200m of the water column are generally absent in the large-scale oxygen distribution of the open Atlantic (see Figure 2a), indicating that such low-oxygen events are associated with isolated transport from the coast or coherent, long-lived eddies combined with biogeochemical oxygen depletion. In the CTD dataset, 74 out of 976 profiles meet this criterion, roughly corresponding to the 10th percentile of all observations.

For the PIRATA mooring data shown in Figure 3, we cannot query the minimum in the upper 200 m, only at the depth of optode measurements, so we initially applied the lowest 10th percentile as a simple criterion. (In other analyses of mooring data throughout the manuscript, the presence of an HBV is further corroborated by velocity data). The PIRATA mooring data shown in Figure 3 were primarily just intended to illustrate variability and low-oxygen events as an overview of a long time series. Nevertheless, we acknowledge that alternative, statistically grounded thresholds should be applied, such as those based on standard deviations or interquartile ranges (IQR). Following this suggestion, we have now applied the IQR method to the PIRATA time series, which confirms and supports the identification of low-oxygen events while providing a more robust statistical definition of outliers. We changed figure 3 to:

[Figure]

And changed the text in the manuscript accordingly to:

L464: *"A low-DO extreme event is defined based on the interquartile range (IQR) of the respective time series, with events identified as values below the lower quartile minus 1.5 × IQR."*

The number of detected events in the time series changed little, except at 11° N and 80 m depth, where one particularly strong event was observed. So, we adapted L469: *"At 11°N, about one event per year occurs at these depths, and at 80 m only one strong event was detected within seven years."*

**While the focus on anticyclonic eddies (ACEs) is understandable given their higher detection frequency in observations, the manuscript would benefit from a (slightly) more symmetric treatment of cyclonic eddies (CEs). Figures 9 and 10 do a good job of characterizing ACE dynamics and evolution via model output; a similar analysis of a representative CE from the model could be similarly instructive. For instance, this could be used to tie in the discussion around CE instability and decay mechanisms (e.g., interaction with high-PV water, lines 884–893). Including this could both reinforce the contrast between eddy types while also showing their similarities, at least from model output.**

We thank the reviewer for this valuable suggestion. We agree that a similar analysis of cyclonic eddies (CEs) could be informative and would provide additional context regarding CE dynamics, instability, and decay mechanisms. However, performing a comparable in-depth analysis for CEs would effectively constitute a separate study beyond the scope of the current manuscript. In addition, our focus in the manuscript is on the observed high-barocline anticyclonic vortices (HBVs/ACEs), which are much more frequently detected in the dataset. Moreover, we do not consider the model to fully reproduce the statistics and properties of the eddies; the model is primarily used here as a complementary tool to support interpretation of the observational data. A detailed analysis of CEs would therefore rely solely on model output. We therefore chose to maintain the emphasis on ACEs, while noting in the discussion that CEs may exhibit complementary dynamics.

**I also have additional minor comments, labeled with specific line numbers:**

**[L. 45-46]: The authors could cite the recent study from Deutsch et al. (2020) (https://doi.org/10.1038/s41586-020-2721-y). That study convincingly shows that temperature and O₂ shape the biogeography of marine organisms.**

Thank you for pointing that out. The study is indeed very relevant, and we have now mentioned it there.

**[L. 117]: The authors could introduce the acronym 'SCV' here after the first mention of submesoscale coherent vortices.**

We have changed several instances of "SCV" to "HBV" in the text. "Subsurface Coherent Vortices" is now only mentioned once in the introduction, which is why we decided not to abbreviate it.

**[L. 124]: Did the authors mean to write "mesoscale-permitting"?**

We thank the reviewer for pointing this out, that was misleading. The model we use is eddy-rich, and at low latitudes it can resolve submesoscale features, though not fully. To clarify, we have revised the sentence to read:

Line 124: *However, ocean models are often submesoscale "permitting" only, in the sense that the model has sufficient resolution to begin representing submesoscale processes but does not fully resolve them, particularly with increasing distance from the equator.*

**[L. 212]: This makes it seem like authors are only showing WOA oxygen during the model validation, but the authors frequently cite MIMOC data in the text. If MIMOC includes oxygen, please mention that here.**

We thank the reviewer for pointing this out. You are correct - the MIMOC dataset publicly available does not provide dissolved oxygen. This was an oversight in the initial version of the manuscript. All oxygen figures with regard to spatial patterns, including Fig. 2a, were now generated using the World Ocean Atlas 2023 (WOA23). We have corrected the data source accordingly and removed all references to MIMOC in the revised manuscript.

**[L. 219]: CM2.6 only has a resolution of 0.1 degree, is that high enough to resolve HBVs? It may be helpful to briefly discuss the model resolution (both horizontal and vertical) in the context of HBV scales, especially if the model is close to the margin of resolving such structures.**

We thank the reviewer for raising this important point. The GFDL CM2.6 model has a nominal ocean resolution of 0.1°, which corresponds to roughly 10 km in our study region near 10° N. At these low latitudes, CM2.6 is mesoscale eddy-resolving and submesoscale-permitting, resolving only the larger submesoscale features (Hallberg et al., 2013). As shown in our Figure 1, the local first baroclinic Rossby radius of deformation (60–150 km in the area of interest) is resolved with approximately more than six grid cells in the model. However, the resolution is close to the lower limit for explicitly representing HBV-scale vortices, which typically have observed radii of 20–45 km (average ≈ 34 km in our observations). Nevertheless, the model is capable of reproducing coherent, high-baroclinic anticyclones with reasonably realistic horizontal scales, consistent with both our observations and previous studies (e.g., see Fig. 6 or Zhang et al., 2021; Frenger et al., 2018).

As our aim is to use the model primarily as a supporting tool to complement the observations and to provide additional insight into the origin and persistence of the vortices, we consider CM2.6 suitable for this purpose. However, we agree that this limitation should be explicitly mentioned. To address this, we have added the following clarification in the revised manuscript:

Lines 393 and the following: "*With a nominal ocean resolution of 0.1°, CM2.6 is mesoscale eddy-resolving and submesoscale-permitting at low latitudes, capturing only the larger submesoscale vortices. The local Rossby radius of deformation (60-150 km; Fig. 1) in the area is resolved, but smaller eddies are near the lower limit of resolvable scales. However, the model has been shown to simulate low-oxygen mesoscale eddies at latitudes poleward of about 12°*

*(Frenger et al., 2018) and provides a useful framework in this study to complement the observational analysis."*

**[L. 235]: What do the authors mean by "five daily model outputs"? I'm assuming they mean to say the output resolution is every 5-days. Please clarify.**

We meant that we used model output averaged over five-day intervals. We hope this is clearer now in the manuscript.

New Line 247: *Here, we used model output averaged over five-day intervals for the last 20 years of the simulation.*

**[L. 260]: Could the upper OMZ in observations be caused by HBV advection? If so, doesn't that say that the model is not accurately capturing their influence?**

The upper OMZ primarily arises from biological oxygen consumption in the upper water column, modulated by physical transport processes, including HBVs. For example, further north, Schütte et al. (2016b) estimated a reduction of 7-16 µmol kg$^{-1}$ in the depth range of the shallow OMZ due to eddies. HBVs are therefore not the sole cause, but can locally enhance the intensity and position of oxygen minima. We agree that the model may underestimate the effects of HBVs on the upper OMZ, as the eddies in the model tend to be somewhat weaker and do not produce as strong low-oxygen anomalies as observed (e.g., Fig. 6). We have added a sentence in the revised manuscript (Lines 770 and following) to clarify that the model likely underestimates the impact of HBVs on the observed DO distribution.

Line 770: *The model tends to slightly underestimate PV and associated O₂ anomalies, indicating somewhat weaker eddy coherence compared to observations. At the same time, due to reduced dissipation in the circulation model, the lifespan of the eddies is slightly prolonged. Additionally, the MiniBLING model does not fully account for remineralization processes in the mesopelagic zone, which likely leads to an underestimation of oxygen consumption. Taken together, this implies that HBVs in the model appear with weaker anomalies but with an artificially prolonged lifespan, which we consider in our interpretation of the results.*

**[L. 261]: Just a suggestion, but since the authors mention depths deeper than 500m, then panels in Figure 2-f could be extended to at least 700m (the deepest depth mention during the validation).**

This is correct. We mention 700 m in the text as the lower boundary of the deep OMZ. However, for the manuscript and the focus of this study, the upper 200-300 m are most relevant. We aimed to focus on this region. Extending the axis further would reduce the visibility of key details in the figures. We therefore prefer to keep it as is. This choice ensures that key structures and variability in the upper OMZ (where the HBVs are) are clearly visible.

**[Section 3.1.1 - 3.1.1 & 3.2]: While very useful to include, these sections could be moved to a Supplementary material. The methods section is quite long as currently presented, and these sections broadly introduce standard oceanographic methodologies (e.g., methods introduced in physical oceanography textbooks). However, if manuscript length is not a**

**concern, feel free to keep them in since they are very useful to frequently reference during discussion of results (Section 4).**

We thank the reviewer for the suggestion. We carefully considered moving Sections 3.1.1–3.2 to the Supplementary Material (we also discussed that during the initial manuscript preparation). However, after internal discussion among the authors, we decided to keep these sections in the main text. We have no length restrictions, and we believe that including the detailed methodological descriptions is beneficial. While these sections are indeed extensive, many standard procedures (e.g., modal decomposition and fitting) are not always sufficiently or clearly described in the literature. Providing a step-by-step explanation within the manuscript helps ensure reproducibility and clarity. We therefore prefer to retain them in the main text.

**[L. 392]: "The horizontal eddy center at each model time step". The authors don't mean the computation time-step here, but the output frequency of the model (5 days?). It could be helpful to clarify this.**

Indeed. We thank the reviewer for pointing out this ambiguity and added: Line 426: *"The horizontal eddy center was determined for each 5-day model output and…"*

**[L. 421]: You could present this additional time-series in the supplementary material.**

We can do that. We have created a supplementary document where all the mooring time series are provided.

**[L: 483]: Can the authors speculate on what is driving the events not linked to subsurface eddies (#5, #6, #8-10)?**

Events #5 and #6 are most likely associated with subsurface eddies, based on their spatial and temporal characteristics. However, due to the lack of velocity data for these cases, we cannot conclusively demonstrate the eddy structure, which is why we refrained from making a definitive claim. For events #8–10, the interpretation is less clear. While one could speculate that an HBV core may have missed the mooring, and we are possibly only observing the southern edge of an HBV. In general, we chose to take a conservative approach, avoiding strong interpretations where the available data do not allow for robust confirmation.

**[L. 523]: Just a suggestion, but the authors could use sea-surface height anomaly products (e.g. Satellite or re-analysis), mapped to the location of the mooring, to determine if there was a pronounced surface signature of these events. That could help determine if the feature is driven by surface-intensified ACEs (if strongly positive) or subsurface-intensified ACEs (if no or weak signature). The authors choose to do this for CTD profiles around L. 585, so why not extend this here? If their arguments from L. 585-588 hold, then mention this for the mooring data as well.**

We thank the reviewer for this suggestion. We indeed examined satellite products extensively at the mooring locations. While very weak signals could occasionally be associated with the events, they were neither unambiguous nor as pronounced as those observed further north near the Cape Verde region. Tracking these features via satellite was therefore not feasible. This is actually a key point of our study: the smaller, southern HBVs cannot be reliably detected from satellite data.

Regarding the text around L. 523, we acknowledge that it was somewhat unclear. There, we intended to convey that the mooring data do not capture the complete vertical structure of the eddies up to the surface. The lack of a surface signature is confirmed by the satellite data. We added a sentence in the manuscript after that paragraph to make this point clearer:

L589: *"Notably, none of these vortices exhibited a clear surface signature in satellite data that could be unambiguously associated with the subsurface features."*

**[L. 696]: In this section, the references for specific Figure 6 panels are incorrect. Please update them.**

Thank you. This is done.

**[L. 748]: This should be referencing Figure 9, not Figure 8.**

Thank you. This is done.

**[L. 862]: Why do the authors report the model having Ro of roughly 0.4 when the time-series in Figure 10 clearly shows lower values near 0.1?**

This is correct: the value of ~0.4 from the model referred to a specific eddy snapshot shown in Figures 6g-l. The time series analysis presented in Figure 10 provides a more representative estimate of the model Rossby numbers, which are generally lower, around 0.1-0.2. Accordingly, we have updated the sentence in the manuscript to reflect this.

Line 946: *"Rossby numbers were below 1, with values of approximately 0.3-0.7 estimated from shipboard observations (one eddy crossing is shown in Figure 6; others not shown) and around 0.1-0.4 in the GFDL CM2.6 model simulation (exemplarily shown in Figure 6 and Figure 10e)."*

**Typos (note there were several more, but I forgot to write their locations, so a second read-through is warranted):**

Yes, we did that and corrected several typos.

**[L: 170]: "…additionally a DO sensor…" (an —> a)**

Done

**[L: 424]: There a few typos in this sentence.**

Changed in: Line 434: *As expected, both the DO variability and amplitude of DO anomalies are generally greater at shallower depths (e.g., 80 m), due to more intense near-surface dynamics and elevated background DO concentrations.*

**[L. 914]: Africa.**

Done

References:

Frenger, I., Bianchi, D., Stührenberg, C., Oschlies, A., Dunne, J., Deutsch, C., Galbraith, E., and Schütte, F.: Biogeochemical Role of Subsurface Coherent Eddies in the Ocean: Tracer Cannonballs, Hypoxic Storms, and Microbial Stewpots?, Glob. Biogeochem. Cycle, 32, 226-249, doi:10.1002/2017GB005743, 2018.

Hallberg, R. (2013). Using a resolution function to regulate parameterizations of oceanic mesoscale eddy effects. Ocean Modelling, 72, 92–103. https://doi.org/10.1016/j.ocemod.2013.08.007

Schütte, F., Karstensen, J., Krahmann, G., Hauss, H., Fiedler, B., Brandt, P., Visbeck, M., and Körtzinger, A.: Characterization of "dead-zone" eddies in the eastern tropical North Atlantic, Biogeosciences, 13, 5865-5881, 10.5194/bg-13-5865-2016, 2016b

---

## Author Comment (AC3)

*Response to Reviewer #3*

We sincerely thank Eric Machu for his thoughtful and constructive feedback, as well as for the positive and encouraging overall assessment of our work. We are especially grateful for the recognition of the challenges involved in observing and analyzing high-baroclinic-mode vortices (HBVs), and for acknowledging the value of combining long-term in situ observations with high-resolution modeling. Eric Machu's detailed comments have helped us clarify several aspects of the manuscript and improve its overall quality. We have carefully addressed all suggestions and concerns raised, and we believe that the revised version of the manuscript is significantly improved as a result. Below, we respond point-by-point to each of the reviewer's comments. Reviewer comments are shown in **black**, followed by our responses in **green**. Changes to the manuscript are highlighted in *italics* and described where appropriate.

**Referee #3: Eric Machu**

*Note:* The reviewer comments are in **black**, our responses are in **green**

**General Comments**

**This study convincingly presents the presence of high baroclinic mode vortices (HBVs) in the low latitudes (4°-12°N) of the North Atlantic and their effect on the presence of oxygen-depleted "islands" in its eastern to central part. The study describes the characteristics, origin and temporal evolution of eddies marked by low oxygen concentrations. The study is based on numerous observations acquired in this region over the last 15 years from sea campaigns and 3 moorings. The understanding of the evolution of these structures is completed by the exploitation of a coupled physical-biogeochemical modeling experiment.**

**The manuscript is very well written, organized and illustrated with relevant, high-quality figures. The work as a whole is of good quality and the results deserve to be published, as they shed light on the presence of small-scale dynamic structures, which are very poorly documented and difficult to observe (and to some extent modeled), with consequences for the biogeochemical characteristics of this oceanic region.**

Thank you very much for the kind words and for placing this work in a scientific context.

**A coupled model is used in this work to present the mechanisms by which an HBV is maintained and evolves over several months from the upwelling zone across the Atlantic basin. Given that the model represents oxygen minima much further west than observations show, and that the mean state of the oxygen variable presents a number of biases (notably the absence of shallow OMZ in the east), I would tend to think that the biogeochemical model is partially inadequate, and I wouldn't necessarily have put so much confidence in the respiration rates derived from it. Given the duration of the simulation, I would also have expected to see occurrence statistics for the eddies described in this study from the simulation.**

We thank the reviewer for the comment. We fully acknowledge that the coupled model used here is not perfect and exhibits biases, as is the case for any ocean-biogeochemistry model. In particular, the model tends to slightly underestimate potential vorticity and the associated oxygen anomalies, leading to somewhat weaker eddy coherence compared to observations.

At the same time, due to reduced dissipation in the circulation model, the lifespan of the eddies is slightly prolonged. Additionally, the MiniBLING model does not fully account for remineralization processes in the mesopelagic zone, likely leading to an underestimation of oxygen consumption. Taken together, this implies that HBVs in the model appear with weaker anomalies but with an artificially prolonged lifespan, which we explicitly consider in our interpretation of the results now.

New Line 770: *The model tends to slightly underestimate PV and the associated $O_2$ anomalies, indicating somewhat weaker eddy coherence compared to observations. At the same time, due to reduced dissipation in the circulation model, we expect lifespans of the eddies to be slightly prolonged. Additionally, the MiniBLING model does not fully account for remineralization processes in the mesopelagic zone associated with HBVs, which likely leads to an underestimation of oxygen consumption. Taken together, this implies that HBVs in the model appear with weaker anomalies but with an artificially prolonged lifespan.*

Despite these limitations, our intention within the manuscript was to use the model as a complementary tool to the observational dataset, providing further insight into the origin and evolution of the eddies. In the revised manuscript, we strengthened the observational support for HBVs by including nitrate as an additional tracer: the new Figure 8b shows observed oxygen concentrations, the depth of the oxygen minimum, and the corresponding nitrate profiles from CTD casts taken inside and outside of low-oxygen events. These data reveal substantially lower oxygen concentrations between 80-250 m, accompanied by elevated nitrate levels inside HBVs, consistent with ongoing biological remineralization and thus "older" water. This supports the interpretation that HBVs represent persistent, isolated water masses rather than short-lived anomalies. We believe this addition strengthens the observational evidence and nicely complements the model-based findings, as discussed at the end of Section 4.5 *Source water of high-barocline vortices* as follows:

New Line 711 and following: *"To further support the persistence and longevity of HBVs, we analyzed CTD observations of oxygen and nitrate inside and outside of low-oxygen events. Fig. 8b shows the median oxygen profiles for CTD casts with a minimum in the upper 200 m of the water column below 60 µmol/kg (blue curve) and those above 60 µmol/kg (orange curves). Mixed layer oxygen concentrations for both cases indicate increased near-surface biological productivity of HBVs compared to outside of HBVs. The red stars indicate the depths of the observed oxygen minima clustering between 80 to 120m depth. Corresponding nitrate profiles are shown in turquoise (<60 µmol/kg oxygen) and yellow (>60 µmol/kg oxygen). The results reveal substantially lower oxygen concentrations between 80 - 250 m inside HBVs, accompanied by elevated nitrate levels, consistent with enhanced accumulated biological remineralization due to enhanced productivity and/or "older" water. This observational evidence indicates that HBVs consist of persistent, isolated water masses rather than short-lived anomalies."*

In addition, regarding the model itself, we now provide a more detailed description of its resolution and capabilities with regard to the observed eddies:

New Lines 393 and the following: *"With a nominal ocean resolution of 0.1°, CM2.6 is mesoscale eddy-resolving and submesoscale-permitting at low latitudes, capturing only the larger submesoscale vortices. The local Rossby radius of deformation (60-150 km; Fig. 1) in*

*the area is resolved, but smaller eddies are near the lower limit of resolvable scales. However, the model has been shown to simulate low-oxygen mesoscale eddies at latitudes poleward of about 12° (Frenger et al., 2018) and provides a useful framework in this study to complement the observational analysis."*

Finally, we would like to emphasize again that in this study, with regard to the model, we focus on case studies of modeled submesoscale eddies to demonstrate plausible dynamics and mechanisms rather than to derive quantitative estimates of respiration rates or oxygen concentrations. We leave quantitative statistics for future dedicated studies, that perhaps could be carried out using GLORYS 1/12° reanalysis. The model results here are used as illustrative examples to contextualize our observational data and facilitate their interpretation and mechanistic insight. All quantitative conclusions regarding HBV occurrence, longevity, and impact rely on observational evidence. We hope that these clarifications and additions in the revised manuscript now better convey our reasoning and the role of the model in supporting, but not replacing, the observational evidence.

**Finally, the study could have been strengthened, particularly the occurrence statistics for this type of vortex, by integrating data from the ARGO profilers. The association of events with vortices could not have been complete, given that currents are not measured, but the study of these numerous profiles (including the depth of isopycnes) could have reinforced the information on the frequency of low-oxygen events in the subsurface waters and hence on their role in the biogeochemistry of this oceanic region.**

We have conducted similar analyses using ARGO floats for eddies north of 12°N, around the Cape Verde region and the Mauritanian/Senegal upwelling zone (Schütte et al., 2016a). We attempted to perform similar analyses in the near-equatorial region; however, we encountered several obstacles that ultimately prevented us from integrating these datasets into the manuscript. First, there is almost no supporting satellite signal (contrasting to the regions further north), as the eddies in the near-equatorial region appear to have little or very weak surface expression, hindering co-location. As you noted, we also lack velocity measurements and oxygen observations (only few ARGO floats provide oxygen), which prevents direct attribution to low-oxygen eddies. In contrast, for the CTD-based assignments, we were able to use direct oxygen measurements and shipboard velocities, which we consider much more reliable and unambiguous. One could, as you suggested, attempt to use ARGO-derived isopycnal displacements; we did explore this, but this approach is prone to noise and confounding processes (other factors displacing the isopycnals), we could not reliably link such displacements to low-oxygen events, and were not satisfied with the results. For these reasons, we ultimately decided not to include ARGO-based analyses at this point, and we believe that doing so would not change the conclusions of our study (as we have a substantial amount of CTD Data, and in addition, long-term mooring datasets for statements about the frequency of low-oxygen events.

**Specific comments**

**Various places in the manuscript mention the link between HBVs/oxygen depletion and biogeochemistry, biodiversity or pelagic fish:**

**L. 71-72: "*Such eddies have a severe impact on biogeochemical processes and organisms*"**

**L. 123-124: "*HBVs can play a crucial role in biogeochemical cycles and marine ecosystems*"**

**L. 952-953: "*which may impact on pelagic fish, biodiversity and biogeochemical cycles*"**

**These structures contribute to the definition of the mean state of the oxygen variable, but it seems difficult to say more, given that the study does not focus on statistics or trends associated with these HBVs. We are talking here about a few events per year, biogeochemical processes and a specific response from planktonic communities are probably associated with them, but to speak of severe impact on organisms sounds rather alarmist and it's hard to imagine a determining role for the presence of these structures for higher organisms over their entire life cycle.**

We thank the reviewer for this helpful comment. We agree that the wording in the manuscript was too strong and may have overstated the broader ecological significance of HBVs. While these vortices can indeed have a regional and short-term impact on biogeochemical conditions and local plankton communities due to their transport and oxygen-depleting effects, our study does not provide direct evidence for long-term or large-scale impacts on higher trophic levels. We have therefore softened the statements in the abstract and discussion to better reflect the scope of our findings. The revised text now emphasizes that HBVs *may locally modulate* biogeochemical processes and ecosystem conditions, rather than exerting a severe or determining influence.

Line 71-72: "*Such eddies can locally modulate biogeochemical processes and influence marine organisms.*"

Line 123-124: "*HBVs may play a role in shaping local biogeochemical conditions and ecosystem variability.*"

Line 952-953: "*which may have localized effects on pelagic habitats, biodiversity, and biogeochemical cycling.*"

**L. 66-67: Could you indicate values or a reference for ambient conditions / respiration rates?**

The ambient respiration rates in the tropical North Atlantic, outside of eddy cores (in median depths of around 80 m), are reported in Schütte et al. (2016b) and amount to approximately $0.04\text{-}0.06\ \mu mol\ kg^{-1}\ d^{-1}$ at similar depths. We have added this information to the manuscript to clarify the comparison with enhanced respiration rates inside the eddies.

Line 66-70: "*Respiration rates in the eddy's interior (at around 80m depth) were found to be substantially increased, with up to 3 to 5 times the values of ambient conditions for the tropical North Atlantic (approximately $0.04\text{-}0.06\ \mu mol\ kg^{-1}\ d^{-1}$), e.g. subsurface intensified anticyclonic eddies (subsurface ACEs): $0.19 \pm 0.08\ \mu mol\ kg^{-1}\ d^{-1}$ and surface intensified cyclonic eddies (CEs): $0.10 \pm 0.12\ \mu mol\ kg^{-1}\ d^{-1}$ ( Schütte et al. 2016b).*"

**L262-263: It would be interesting to mention the main reasons put forward to explain these biases.**

We thank the reviewer for this comment. Indeed, the shallow OMZ in the tropical Atlantic (above 200 m) is not well reproduced by the model, which instead simulates a single OMZ spanning roughly 100 - 600 m. This bias is commonly found in coupled ocean-biogeochemistry models (e.g., Duteil et al., 2014) and is attributed to several factors: insufficient vertical resolution in the upper ocean, which smooths out rapid oxygen depletion; simplified or parameterized remineralization and biological processes that fail to reproduce fast upper-ocean oxygen consumption; and limited representation of physical transport processes such as submesoscale eddies and coastal upwelling, which locally enhance oxygen minima. Additionally, models often overmix oxygen from deeper layers into the upper ocean, further weakening the shallow OMZ signal. This underscores the importance of observational studies such as ours, which provide insights into the shallow oxygen minimum, its connection to low-oxygen events, and the role of high-baroclinic vortices. We have added a brief discussion of these points in the Introduction to further motivate why studies based on direct observations remain essential.

New Line 292: “*This bias is also present in other coupled ocean circulation biogeochemistry models (e.g. Duteil et al., 2014) and can be attributed, among other factors, to the limited representation of physical transport processes such as submesoscale eddies, which locally enhance oxygen minima. Additionally, simplified or parameterized remineralization and biological processes fail to reproduce rapid upper-ocean oxygen consumption. These discrepancies highlight the importance of direct observational studies, such as ours, which provide detailed insights into the shallow oxygen minimum and its connection to low-oxygen events and high-baroclinic vortices, thereby motivating the focus of this study.*”

**L. 696-697: “*we identified a HBV with a low-DO core in the near-equatorial open ocean as exemplarily shown at the position 28°W, 10°N.*”**

**Why haven't you compiled statistics on the occurrence of this type of vortex throughout the simulation?**

As discussed in previous responses, we consider the model useful primarily to illustrate plausible HBV dynamics and mechanisms. Given known biases - such as underestimated eddy intensity, weaker oxygen anomalies, and imperfect representation of shallow OMZs - we refrain from deriving quantitative statistics from it; such statistics would be better obtained from higher-resolution products or future dedicated studies. All quantitative conclusions regarding the occurrence, longevity, and impact of HBVs are therefore based solely on observational data, including CTD sections and PIRATA mooring time series.

**L.765-766: “*DO continuously decreases from 97 µmol/kg to 54 µmol/kg over 300 days which yields an average DO consumption of 0.14 µmol/kg/d*”**

**L. 894-895: “*DO decrease from 92 to 46 µmol/kg over a period of 260 days, yielding a DO consumption of 0.18 µmol/kg/d*”**

**Why calculate two different respiration rates? As mentioned in the general comments, I'm not sure that the model used is relevant for estimating respiration levels, since it does not reproduce depleted structures in the right longitude bands and, on average, fails to reproduce a shallow OMZ signal (and the spatial extent of the OMZ). Observations show the occurrence of low O2 events far to the west of the basin, but in the absence of**

**statistics on the presence of these events in the model, we question the representativeness of the example of HBV studied from the simulation.**

First, we agree that reporting two different respiration rates for slightly different time spans of the modeled HBV is not particularly helpful and can be confusing. Second, we also agree that such specific numerical values derived from the model are particularly questionable. We have therefore removed these two examples from the manuscript and replaced them with a general, qualitative statement.

Line 896: "*Oxygen in the HBVs decreased from roughly 95 µmol kg$^{-1}$ to 50 µmol kg$^{-1}$ over several months, corresponding to an average consumption rate of about 0.16 µmol kg$^{-1}$ d$^{-1}$. This apparent decline should be regarded as a lower limit, as ventilation and mixing processes would partly offset oxygen loss.*"

Line 917: "*The magnitude and timescale of this decrease are broadly consistent with observed low-oxygen events in the region, though specific rates from the model should be interpreted cautiously.*"

Line 1046: "*During the lifetime of the simulated anticyclonic HBVs, enhanced respiration within the eddy core contributes to a noticeable decrease in DO over several months. While this trend is qualitatively consistent with observations of low-oxygen events, the model-derived values should be considered indicative rather than quantitatively precise.*"

**Technical corrections**

**L. 83 : correcting « dominace » into « dominance »**

Thank you. Done

**L. 146-147, 167, 170: Homogenize the writing of the different mooring locations (coordinate pairs) for in situ moored, shipboard observations + ocean-biogeochemistry model**

We thank the reviewer for pointing this out. We have now homogenized the writing of all mooring and observation locations throughout the manuscript, including in situ moorings, shipboard observations, and model grid points. All coordinates are now consistently presented in the format °N/°W throughout the text, figures, and tables.

**L. 482: correcting « asciated » into « associated"**

Thank you. Done

**L. 616: correcting "isopyncal" into isopycnal"**

Thank you. Done

**L. 856: correcting "verticies" into "vorticies"**

Thank you. Done

---

## Author Response (AR2)

Dear Editor,

thank you very much for the follow-up and for indicating the remaining technical points. We have now addressed all of them in the revised manuscript:

L.766: corrected "sissolved" to "dissolved".

*Done*

L.956: updated the sentence to "The DO consumption rates found here…".

*Done*

L.963: updated the sentence to "the low-DO eddies described here…".

*Done*

L.1030: correct the DOI.

*The correct DOI for all the Data used in this study is updated: https://doi.pangaea.de/10.1594/PANGAEA.987397*

Figure 8a: removed all MIMOC references and updated the figure accordingly.

Thanks, we missed that. It's been fixed now.

Please let us know if any additional adjustments are needed.

Kind regards,